# Skin Cancer Pathobiology at a Glance: A Focus on Imaging Techniques and Their Potential for Improved Diagnosis and Surveillance in Clinical Cohorts

**DOI:** 10.3390/ijms24021079

**Published:** 2023-01-05

**Authors:** Elena-Georgiana Dobre, Mihaela Surcel, Carolina Constantin, Mihaela Adriana Ilie, Ana Caruntu, Constantin Caruntu, Monica Neagu

**Affiliations:** 1Faculty of Biology, University of Bucharest, Splaiul Independentei 91-95, 050095 Bucharest, Romania; 2Immunology Department, “Victor Babes” National Institute of Pathology, 050096 Bucharest, Romania; 3Department of Pathology, Colentina University Hospital, 020125 Bucharest, Romania; 4Department of Dermatology, Kalmar County Hospital, 391 85 Kalmar, Sweden; 5Department of Oral and Maxillofacial Surgery, “Carol Davila” Central Military Emergency Hospital, 010825 Bucharest, Romania; 6Department of Oral and Maxillofacial Surgery, Faculty of Dental Medicine, “Titu Maiorescu” University, 031593 Bucharest, Romania; 7Department of Physiology, “Carol Davila” University of Medicine and Pharmacy, 050474 Bucharest, Romania; 8Department of Dermatology, “Prof. N.C. Paulescu” National Institute of Diabetes, Nutrition and Metabolic Diseases, 011233 Bucharest, Romania

**Keywords:** squamous cell carcinoma, cutaneous melanoma, reflectance confocal microscopy, optical coherence tomography, magnetic resonance imaging, positron emission tomography

## Abstract

Early diagnosis is essential for completely eradicating skin cancer and maximizing patients’ clinical benefits. Emerging optical imaging modalities such as reflectance confocal microscopy (RCM), optical coherence tomography (OCT), magnetic resonance imaging (MRI), near-infrared (NIR) bioimaging, positron emission tomography (PET), and their combinations provide non-invasive imaging data that may help in the early detection of cutaneous tumors and surgical planning. Hence, they seem appropriate for observing dynamic processes such as blood flow, immune cell activation, and tumor energy metabolism, which may be relevant for disease evolution. This review discusses the latest technological and methodological advances in imaging techniques that may be applied for skin cancer detection and monitoring. In the first instance, we will describe the principle and prospective clinical applications of the most commonly used imaging techniques, highlighting the challenges and opportunities of their implementation in the clinical setting. We will also highlight how imaging techniques may complement the molecular and histological approaches in sharpening the non-invasive skin characterization, laying the ground for more personalized approaches in skin cancer patients.

## 1. Introduction

The skin is the immune organ subjected to the most physical, chemical, and biological agents; therefore, it is the common site of cancer diagnosis [1]. By far, cumulative exposure to solar UV radiation (UVA and UVB) remains the most prominent risk factor for skin cancer since it leads to genomic instability, inflammation, and the emergence of cutaneous tumors in humans, even decades after initial exposure [2,3]. Based on the cell of origin and clinical behavior, cutaneous tumors are subdivided into two main categories: cutaneous melanoma (CM) and non-melanoma skin cancers (NMSC) [4]. CM, which evolves from the malignant transformation of melanocytes, the pigment-producing cells in the dermis, accounts for only 4% of skin cancer cases; hence, if left untreated or caught in a late stage, CMs are challenging to treat and potentially life-threatening, being responsible for more than 75% of skin cancer-related deaths [5]. Therefore, CM’s early diagnosis and treatment are vital to maximizing survival rates [6]. On the other hand, NMSC refers to keratinocyte tumors—basal cell carcinoma (BCC) and squamous cell carcinoma (SCC)—the most commonly diagnosed skin cancers globally, accounting for 2–3 million new cancer cases annually [7]. BCC rarely metastasizes and is generally curable but can cause disfigurement if not detected early. Regarding SCCs, although they are not characterized by high lethality if diagnosed and treated in the early stages, 2–4% of primary SCCs tend to spread to other body parts, with patients carrying a poor prognosis in such cases [8].

Skin cancer dramatically affects the quality of life, as it can be disfiguring or even deadly. However, a significant percentage of skin cancer cases can be successfully treated if detected early [9]. Thus, improved methods for early detection and appropriately targeted interventions have the potential to decrease not only the number of cases diagnosed in advanced stages but also the psychological impact associated with such a diagnosis. Besides causing illness and death, skin cancers exert a significant financial burden on healthcare systems. Cutaneous tumors involve additional costs beyond those related to treatment that are associated with lost workdays and restricted activity days [10]. In the US, it is estimated that an individual who dies of CM loses somewhere around 20 potential years of life compared to 16.6 years for other cancers. Therefore, gaining a deeper understanding of this disease’s biology and improving its early detection may decrease the socioeconomic impact of skin cancer patients’ premature mortality and the psychological implications of such a diagnosis [11].

The visual inspection of suspicious skin lesions with the naked eye and dermoscopy are the primary methods for diagnosing skin cancer. However, these procedures are inaccurate and depend on the dermatologist’s training level [12]. The histological examination, which involves the surgical excision of the tumor, remains the cornerstone of a skin cancer diagnosis. Nonetheless, the biopsy procedure is highly invasive and results in pain, anxiety, and cosmetic defects. In addition, it has a long turnaround time, resulting in a lag between the initial assessment and definitive treatment. As an added complication, concerning levels of inter-observer and intra-observer variability have been reported among pathologists when analyzing biopsies of neoplastic skin conditions, yielding estimates that around 17% of the diagnoses for melanocytic lesions in the US are incorrect [13]. This has prompted clinicians and scientists to look toward more accurate and in-real-time exploratory assays for skin cancer diagnosis, such as imaging techniques. A bourgeoning body of data highlights that noninvasive imaging may represent a promising strategy for early cancer detection while preventing unnecessary skin biopsies in suspected patients [14].

The latest advances in technologies and artificial intelligence (AI) have opened up new opportunities for visualizing tissue architecture or exploring the molecular features of cells at unprecedented levels of detail. Consequently, many anatomical and molecular imaging techniques have been developed and employed to interrogate different types of malignant lesions [15]. Overall, the use of non-invasive, high-resolution techniques for the anatomical imaging of skin cancers or related cutaneous lesions significantly increased the diagnostic accuracy for these highly problematic pathologies [16]. The most frequently used techniques with anatomical precision include confocal laser scanning microscopy (CLSM), optical coherence tomography (OCT), multiphoton microscopy (MPM), high-frequency ultrasound (HFUS), terahertz pulsed imaging (TPI), and magnetic resonance imaging (MRI). Although they can be beneficial in many settings, anatomical imaging alone is insufficient to guide diagnosis and therapy since the pathological changes may occur at the molecular level long before any anatomical differences can be detected [17]. Molecular imaging techniques such as single photon emission computed tomography (SPECT/CT) and positron emission tomography (PET) introduce molecular imaging agents (probes) to determine the expression of indicative molecular markers at different stages of disease and therefore are actively employed for skin cancer detection and monitoring. The signaling pathways and molecular targets that may be harnessed for such purposes include glucose metabolism [18], integrin α_v_β_3_ [19], melanocortin-1 receptor (MC1R) [20], PD-1/PD-L1 axis [21], and several other biochemical and molecular markers.

Although the advances we witnessed in recent years in imaging techniques were genuinely groundbreaking, the use of imaging techniques can often be associated with several potential harms, such as overdiagnosis and overtreatment, false diagnosis, and radiation risks [22]. Overdiagnosis may occur when imaging detects an asymptomatic disease that would not have become clinically apparent over an individual’s lifetime [23]. That is, overdiagnosis highlights the inability of imaging techniques to differentiate between lesions that have an indolent course from those that would have an aggressive course [24]. False positives are suspicious findings of cancer presence that are not tumors, confirmed by histopathology, imaging, and clinical follow-up tests [25]. In contrast, false negatives are results indicating that a person does not have a particular disease when the person does have the condition. However, overdiagnosis and false diagnosis can lead to severe harm, including psychological distress, consequences of subsequent testing (including invasive tests), treatment exposure and follow-up, and financial impact on the individual and society [26]. In addition, false negatives may result in delayed or a lack of supportive treatment and may have a negative impact on public confidence on screening. Therefore, radiologists need to be knowledgeable about these issues to educate patients and inform colleagues about the harms and appropriate use of imaging-based screening. Finally, image interpretation has large inter-observer and intra-observer variability, which may significantly influence measurements in cancer patients, resulting in erroneous interpretations [16]. Ongoing efforts to standardize the terminology, technique, interpretation, reporting, and data collection in functional imaging are expected to help solve these limitations in the future.

This review discusses the latest technological and methodological advances in imaging techniques that may be applied for skin cancer detection and monitoring. In the first instance, we will describe the principle and prospective clinical applications of the various imaging techniques, highlighting the challenges and opportunities of their implementation in the clinical setting. We will also highlight how imaging techniques may complement the molecular and histological approaches in sharpening the non-invasive skin characterization, creating the potential to open new avenues for personalized care in skin cancer patients.

## 2. Anatomical Imaging Techniques

Anatomical imaging techniques are high-resolution techniques aiming at providing a detailed anatomic and physiologic picture of different organs and tissues within the body for diagnostic, treatment, and monitoring purposes [17]. Each of these anatomical imaging techniques has its strengths and disadvantages. For instance, reflectance confocal microscopy (RCM) is a high-resolution imaging technique with remarkable labeling capabilities but with limited penetration depth [27]. In contrast, HFUS, OCT, and MRI can precisely acquire the entire lesion thickness, being ideal for cross-sectional tissue imaging; however, they lack cellular resolution, causing the identification of malignant lesions to be extremely difficult [17]. RCM and OCT remain the most promising alternatives to invasive tissue biopsies, being ideal for delineating the tumor margins and monitoring therapeutic responses [16]. HFUS may also evaluate tumor margins or depth and monitor patients after exposure to specific therapeutic algorithms [28]. THz radiation, which is strongly attenuated by water and sensitive to water content, proved to be effective in the diagnosis of human neoplasms (as tumorigenesis relies on an increase in the blood microvasculature and, thus, an increase in the tissue water content) and for evaluating the drug spreading into the skin [29]. Moreover, despite its small penetration depth, THz radiation may be harnessed for targeted DNA demethylation, serving as a potent epigenetic inhibitor in the management of cutaneous malignancies [30]. Finally, regardless of its low resolution, MRI may be beneficial in assessing the configuration and intra-tumor characteristics, such as homogeneity, the extent of local invasion, cyst formation, and hemorrhage, holding a promise for more optimized and individualized therapeutic approaches for cutaneous cancer patients [31].

Although further research is needed to improve the accuracy of anatomical imaging tools, their importance in skin cancer management remains undeniable. Imaging can provide insights into an intact system, being much more relevant and reliable than conventional tissue biopsies or in vitro assays. Hence, it provides more statistically relevant information since longitudinal investigations can be conducted on the same subjects, enabling malignant tumor discrimination from other benign entities, improved patient stratification, individualized anti-cancer treatment follow-up, and dose optimization [17].

This section presents the principle and current applications of anatomic imaging techniques in dermato-oncology, emphasizing the latest advances in the field. The main strengths and limitations of using those imaging tools in skin cancer patients will also be highlighted.

### 2.1. Confocal Laser Scanning Microscopy (CLSM)

Confocal laser scanning microscopy (CLSM) is one of the most versatile and accessible anatomical imaging techniques. It allows the visualization of tissue structures at a resolution similar to histological examinations, thus enabling clinicians to perform an “optical biopsy” in a less invasive fashion [32]. CLSM may be performed in either fluorescence or reflectance mode, depending on the source of contrast imaging [33]. Fluorescence confocal microscopy (FCM), which involves the administration of fluorescent compounds to generate contrast, has been used predominantly for ex vivo studies, showing promising results in examining and diagnosing skin lesions. In contrast, reflectance confocal microscopy (RCM), which exploits the differences in the refractive indices of cellular structures, is commonly used for the in vivo examination of tissues. RCM proved to be a valuable tool for scanning the entire lesion and noninvasively determining the most diagnostically and prognostically significant area to biopsy, which may significantly reduce the risk of sampling error and false-negative rates owing to the increased heterogeneity of the lesions [27].

In recent years, the role of CLSM has become pivotal for examining tissue structures, which is why it has been included as a standard in many commercial microscopes. Here, we will briefly present the latest findings regarding the use of CLSM in skin cancer management.

#### 2.1.1. Reflectance Confocal Microscopy (RCM)

RCM employs a near-infrared low-power laser light beam directed to penetrate the target skin surfaces. Subsequently, the light beam is reflected and filtered by a pinhole to remove any other light interference from the targeted focal area. Henceforth, both the objective lens focus point and the focus point where the aperture is placed align to generate a coincidence of these two focal planes, allowing imaging at a remarkably high resolution (0.5–1 µm) [34]. Each skin component has a particular refractive index and RCM uses this index to generate black and white images representative of the structures aimed at being investigated. Notably, due to their high refractive indexes, constituents such as melanin, keratin, collagen, and hemoglobin backscatter the light and appear brighter than other tissue structures with lower refractive indexes. The images are mainly taken horizontally, parallel to the skin’s surface, but dysregulated lesions can also be seen in the oblique plane, thus complementing the vertical assessment allowed by histopathology [35]. There are at least two versions of RCM devices. Wide-probe RCM should be fixed on the suspicious lesions on flat skin areas and can create up to 8 × 8 mm^2^-sized mosaic images. In contrast, handled RCM does not require fixation on the skin and can examine lesions located on curved areas such as the ears. Handled RCM covers an 0.75 × 0.75-mm^2^-sized area and has a shorter imaging time than the wide-probe RCM version [16]. Due to its high resolution, RCM may be applied to all types of skin cancers with increased specificity and sensitivity; nonetheless, it presents a limited depth of penetration of about 200 µm, which may lead to false-negative results for tumors located below the papillary dermis [16].

BCC is the most common skin cancer that, in as many as 80% of patients, develops in the head/neck region, often in the absence of pre-cancerous lesions. Even if UV exposure is considered the main carcinogen, there are numerous other risk factors possibly involved in its development and progression [36,37,38,39]. BCC rarely metastasizes but frequently shows local invasion and tissue destruction, thus resulting in high morbidity rates among the affected individuals [40].

RCM has proved helpful in assessing the key features of BCCs (see Figure 1), such as telangiectasia and convoluted vessels, tumor nests/islands, palisading (peripheral basaloid nuclei disposed perpendicularly to the axis of the tumor island), and clefting, being ideal for use in BCC diagnosis [41]. At the same time, RCM allows for visualizing the prominent and twisted blood vessels with heavy leukocyte traffic within the dermis and the inflammatory cells surrounding the tumor nests [42]. A systematic review on diagnostic accuracy of RCM in BCC reported a sensitivity of 97% and a specificity of 93% [43]. Moreover, this technology seems to provide vital information for delineating the edges of tumor lesions before BCCs surgical excision [44]. A bourgeoning body of evidence also highlights that RCM offers the advantage of monitoring the therapeutic response after surgical and non-surgical therapies, such as photodynamic therapy (PDT) [45], radiation therapy [46], laser therapy [47], cryotherapy [48], and oral Hedgehog inhibitors [49], eliminating the need for highly invasive, serial tissue biopsies in BCC patients. In addition, several other studies have shown that RCM can be used successfully to identify BCC subtypes [50,51]; however, a recent randomized controlled multicenter study has demonstrated that biopsy outperformed RCM in diagnosing and differentiating subtypes of BCC [52]. Sensitivity for BCC diagnosis was increased and similar for both methods (RCM 99.0% vs. biopsy 99.0%; *p* = 1.0). Nonetheless, specificity for BCC diagnosis was lower for RCM (59.1% vs. 100 %; *p* < 0·001) compared with tissue biopsy and remained inferior to the specificity rates reported in previous studies. Tissue biopsy also proved more accurate in identifying the BCCs with more aggressive clinical behaviors, which is essential for patient risk stratification and treatment selection in the clinical setting [52].

SCC is the second most common cutaneous malignancy after BCC, with an increasing incidence worldwide. SCC accounts for most NMSC-related metastatic cases; therefore, the early recognition and treatment of SCC is essential for preventing neoplastic progression and patient death [8]. Recently, it has been shown that RCM can be successfully used to monitor actinic keratoses (AK), which are precancerous skin lesions that can progress to SCC in 20% of cases [53]. RCM can identify several key features of AK, such as parakeratosis (the presence of highly refractile, round nucleated cells in the stratum corneum) and hyperkeratosis (an increase in thickness of the stratum corneum seen as a refractile amorphous material), which may be critical in AK diagnosis [54]. In addition, RCM may be a useful response monitoring tool for AKs undergoing cryotherapy and PDT [55]. At the same time, RCM proved efficient in evaluating those features that help diagnose cutaneous SCC (see Figure 2). The presence of architectural disarray (an atypical honeycombed pattern with pleomorphic and atypical keratinocytes) in the stratum granulosum alongside architectural disarray in the spinous layer and nest-like structures in the dermis have been reported as the most important hallmarks to distinguish SCC from AK [56]. Recent evidence has suggested that RCM can also be used for the in vivo differentiation between premalignant lesions and SCC located on the lower lip [57,58]. When the thickness of the lesion allows, RCM may also enable the exploration of the dermo-epidermal junctions, in which the dermal papillae appear elongated with intricate blood vessel networks inside them [33]. Hence, a systematic meta-analysis of twenty-five studies reported that the sensitivity and specificity of RCM for the diagnosis of AK, SCC in situ, invasive SCC, and keratoacanthoma (KA) were 79–100% and 78–100%, respectively [59]. Notably, the effectiveness of RCM was limited in infiltrative lesions due to its low penetration power [60]. Further complicating this scenario, residual SCC detection after Mohs surgery often became hindered by non-reflecting features of keratinization, which is why RCM imaging techniques should be used with precaution in the clinical management of SCCs [61].

CM is the most lethal skin cancer among all cutaneous malignancies, particularly if diagnosed at an advanced stage. Regarding CM, RCM has proven extremely useful in detecting and diagnosing melanocytic tumors (see Figure 3), with sensitivity and specificity rates ranging between 92–93% and 70–76%, respectively, among different studies [62,63]. RCM has also been used successfully to detect amelanotic melanoma with poor dermoscopic features. However, the most significant advantage comes from coupling RCM with dermatoscopy, as this procedure reduced by half the number of biopsies needed to diagnose melanocytic tumors compared to dermoscopy alone [64,65]. Last but not least, it should be noted that RCM can be used in CM for the evaluation of surgical margins [66] but also for non-invasive monitoring purposes in patients treated with systemic [67], radiation, or topical therapies [68]. According to the latest scientific reports, the sensitivity of RCM in estimating the risk of recurrence of CM is higher than that of dermoscopy [69].

The current literature highlights that RCM features may help distinguish between melanocytic and non-melanocytic skin tumors when integrated into univariate and multivariate analyses alongside other clinical variables. RCM studies have shown that melanoma cells are polymorphic, roundish, or with branching dendrites and usually have a bright appearance. At the same time, benign nevi presented monomorphic, radiant cells, round to oval in shape [33]. Junctional and dermal nevus cell nests were linked with benign nevi occurrence, whereas an irregular melanocytic cell architecture was depicted in melanoma. Keratinocyte cell borders were also poorly defined or absent in melanomas. Yet, the horizontal optical sections in RCM enabled a better visualization of melanocyte morphology than the classical histological hematoxylin and eosin stain (H&E) [33]. It is also worth mentioning that by correlating RCM-observed cell morphology with histopathological type, Pellacani et al. identified four distinct melanoma subtypes: dendritic cell, round cell, dermal nest, and combined-type melanomas, each with different tumor and biological characteristics [70]. A recent study by the same team highlighted that each RCM melanoma subtype expresses a specific molecular profile and biological behavior in vitro [71]. Ki-67, MERTK, nestin, and several stemness markers, notorious for their association with highly invasive tumor phenotypes, were more frequently identified in combined-type and dermal nest melanomas than in dendritic cell and round cell melanomas. These findings were further confirmed in multicellular tumor spheroids. In addition, the highly aggressive dermal nest melanoma subtype, located predominantly in the dermis, displayed a unique molecular signature, encompassing biomarkers associated with ECM remodeling, angiogenesis, inflammation, and cancer cell stemness [71].

Although highly effective in analyzing tissue morphology, RCM was not initially approved for routine clinical practice in the United States. The main obstacles to its widespread use and testing were the high cost, expertise required for RCM image analysis, and limited training opportunities. In 2008, RCM gained FDA approval to be used by physicians in the early detection of skin cancers to reduce the frequency of unnecessary and highly invasive tissue biopsies [72]. Eight years later, the RCM examination received a Current Procedural Terminology (CPT) code from the American Medical Association, which enabled the reimbursements of the examinations performed and leveraged RCM use in the clinical setting [73].

As in vivo RCM is gaining momentum in dermatology, numerous deep-learning technologies have been developed in recent years to increase the accuracy and expand the application of RCM in the clinical setting. Machine learning-based algorithms are helpful in reducing the number of RCM artifacts the operator has to view, shortening evaluation times, and decreasing the number of patient visits to the clinic [74]. In addition, certain AI algorithms may enable the conversion of gray-scale RCM images into H&E-like color images, reducing the number of invasive skin biopsies necessary for diagnosing a cutaneous tumor. For instance, Li et al. performed virtual histology of in vivo, label-free RCM images of normal skin structure, BCC, and melanocytic nevi with pigmented melanocytes and found that the identified features overlap with the conventional histological findings of the same excised tissue [14]. Therefore, the application of AI algorithms in RCM can revolutionize the clinical management of skin cancer, enabling a more rapid and accurate diagnosis of malignant skin neoplasms and reducing unnecessary skin biopsies in the affected patients.

#### 2.1.2. Fluorescence Confocal Microscopy (FCM)

FCM relies on the fact that when a substance is excited by a laser with a suitable wavelength, its molecules absorb energy, move from the ground state to an excited one, and hence generate fluorescence with a longer wavelength when transitioning back to the ground state [75]. This fluorescence emission may arise from endogenous fluorophores (e.g., lipofuscin or NAD(P)H) or special fluorescent dyes (e.g., acridine orange, methylene, toluidine, Nile blue, and patent blue V) that have been applied to targeted tissues following a specific staining protocol. The specificity of fluorescence staining approaches ranges from the general depiction of the skin structure to the very specific staining of a molecular target within the cells [76]. FCM also offers exceptional labeling and multiplexing possibilities; so that by employing combinations of fluorescent dyes with different emission characteristics, FCM enables the simultaneous visualization of multiple tissue components of interest. However, it should be noted that such investigations are primarily limited to tissue sections with a maximum thickness of ≈10 µm, as the multiple labeled structures may overlap in more dense tissues, leading to ambiguous results and misinterpretation [76]. Yet, the most current applications of the FCM include the in “real-time” pathological examination of freshly excised or frozen specimens for diagnostic purposes and tumor margin assessment in Mohs surgery. FCM provides optical sectioning of approximately 1.5 µm and a resolution of roughly 0.4 µm [77].

The majority of studies conducted with FCM in dermato-oncology have been performed on BCCs. As the backscattered contrast in RCM remains an important challenge for accurate skin cancer diagnosis, FCM has recently emerged as the preferred confocal microscopy method for BCC diagnosis at the bedside. Due to the weak backscatter of intranuclear chromatin, dense nuclear BCCs appear dark to the surrounding dermis; hence, those tumors are easily confoundable with normal dermal structures in RCM. However, this limitation can be overcome by coupling FCM with exogenous molecular-targeted fluorescence nuclear contrast agents [78]. Using fluorescent dyes for staining the skin or labeling different structures of interest enhances the optical sectioning capabilities of the confocal system, allowing a detailed examination of the skin structure and precise localization of the cells or their constituents within an intact 3D environment. Nonetheless, regarding ex vivo tissue biopsies, there are still several concerns on how to preserve the properties of those tissues for as much time as possible when preparing them FCM for examination. Therefore, it is absolutely essential to maintain the integrity of the tissue of interest to enable the 3D visualization of skin samples with increased accuracy and resolution [78].

Acridine orange is the most commonly used fluorescent agent in FCM. In one of the pioneering studies in the field, Gareau et al. exploited the ability of FCM to quickly and accurately detect BCC in thick and fresh surgical BCC excisions. Using acridine orange as a fluorescent contrast agent, they developed a protocol that required 9 min in contrast to the 20–45 min needed to prepare frozen histology samples, providing an alternative toward rapid surgical pathology-at-the-bedside to expedite and guide surgery in BCC. Notably, the obtained confocal mosaics enabled the detection of BCCs with an overall sensitivity of 96.6%, specificity of 89.2%, positive predictive value of 93.0%, and negative predictive value of 94.7% [79]. Nonetheless, Ruini et al. have recently highlighted that either on fresh or frozen tissues, FCM results should be interpreted with care due to the increased prevalence of artifacts [80]. The authors examined the diagnostic accuracy of FCM on BCC tissues stained with acridine orange and digitally colored to simulate H&E dyes and found that, compared to fresh tissue, frozen tissue presented a higher prevalence of artifacts (more than 61% of BCC frozen tissues displayed artifacts). Yet, they concluded that FCM in overlap mode with digital staining accounted for a more accurate evaluation of images when compared to the earlier generation of FCM that offered FCM alone, highlighting the need for more complex AI algorithms to strengthen FCM performances in the clinical setting [80].

Besides acridine orange, the cyanine, rhodamine, fluorescein, and boron (III) dipyrromethenates (BODIPYs) are other promising organic dyes for imaging bio-applications that act as luminophores in the composition of bio-conjugates. Recently, Sahu et al. have evaluated the clinical efficacy of FCM and PARPi-FL, an exogenous nuclear poly-(ADP-ribose) polymerase (PARP1)-targeted BODIPY in 95 fresh-discarded BCC surgical specimens [78]. PARP1 is overexpressed in various tumor types and has an essential role in cancer cell proliferation, angiogenesis, and metastasis due to its effects on regulation of multiple signaling pathways. The authors concluded that images acquired with combined FCM and RCM showed higher diagnostic accuracy for BCCs than RCM alone. PARP-immunohistochemistry (IHC) analysis confirmed the FCM imaging findings, which emphasized consistently higher PARP expression in all BCC subtypes versus normal skin structures. Due to its higher intra-nuclear accumulation in tumors, rapid tissue permeation, safety profiles, and detectability deeper in tissues, PARPi-FL, shows promising potential for FCM in the diagnosis of skin cancer patients [78]. Furthermore, it should be noted that BODIPYs can also be conjugated drugs (such as Olaparib), causing the fluorophore, which functions primarily as a biomarker, to exert therapeutic effects on tumor cells as well [78,81]. In this way, researchers and clinicians can monitor in real-time the drug distribution into the skin and the therapeutic effects exerted by the administered compounds, overcoming the obstacles imposed by the non-fluorescent nature of systemic anti-neoplastic therapies. Those fluorescent drug conjugates act similarly to photodynamic therapy (PDT) agents; briefly, they can be absorbed exclusively by the malignant cells, where, under the influence of the light, they are further metabolized in the form of a cytotoxic drug and a BODIPY. Therefore, the benefits of using BODIPY in FCM for in vivo diagnosis and monitoring of skin cancers, as well as for therapeutic interventions, are outstanding and can be exploited for improved clinical care in the oncological setting [81].

Besides acridine orange and organic dyes, many other exogenous fluorophores (e.g., fluorescent proteins, nanoparticles, quantum dots) may be exploited to capture molecular details relevant to disease evolution, prognosis, and novel therapeutic interventions in (skin) cancer models [82]. One of the most promising approaches to assessing the vasculature relies on the use of Hoechst (Hoechst-33342, Invitrogen, Waltham, MA, USA), a small 615 Da molecule, which penetrates rapidly into the tumor tissue and is internalized within the nucleus of stromal cells [83]. Alternative approaches employ fluorescently labeled dextrans available in different sizes, which extravasate in the tumor mass and are eliminated from the bloodstream within a few hours [83]. Polyethylene glycol (PEG)ylated nanoparticles may also be administered if presence in the blood for more hours is desired. Seynhaeve and ten Hagen highlighted that the injection of PEGylated long-circulating nanoparticles (PEG-NP) labeled with a red fluorescent marker lissamine-rhodamine-phosphatidylethanolamine (Rhodamine-PE) or far-red fluorescent marker dioctadecyl tetramethylindotricarbocyanine perchlorate (DiD) may enable the 4D dissection of tumor vasculature and endothelial cell-pericyte association at the molecular level in mouse models of skin cancer [82]. The dynamic interplay between pericytes and endothelial cells is at the basis of vascular physiology and, indeed, tumor angiogenesis. Therefore, targeting the tumor-associated vasculature represents a valuable strategy in cancer therapy in which these two cell types form the main focus as a clinical target. Notably, when integrated with whole-mount staining, FCM enabled the re-evaluation of endothelial cell-pericyte association without compromising spatial positions in the targeted tumor area and allowed for the identification of different molecular subtypes of vascular cells within the tissues, which may be of great interest for the design of novel anti-cancer targeted therapies [82]. Finally, fluorescently labeled chemotherapeutic agents or therapeutic nanoparticles may be injected to assess treatment effects in living organisms, enabling more comprehensive pharmacological and pharmacokinetic studies over some compounds [84]. Therefore, FCM applications are broad, diverse, and hold promise for improved diagnostic, monitoring, and therapeutic approaches in (skin) cancer management.

### 2.2. Multiphoton Microscopy (MPM)

Multiphoton microscopy (MPM), based on the excitation and detection of nonlinear optical signals from tissues and cells, is another important technique that may be used for the noninvasive imaging of biological tissues, especially the thick ones [1]. There are at least two variants of MPM, such as two-photon excited fluorescence (TPEF) and second harmonic generation (SHG), which may be used for investigating tissue morphology, functionality, and biochemical composition both in vivo and ex vivo [2].

TPEF induces the transition of molecules into excited electronic states by the simultaneous absorption of two low-energy photons in a third-order nonlinear process. Similarly with CLSM, TPEF allows the visualization of both endogenous (nicotinamide adenine dinucleotide (NADH), flavin adenine dinucleotide (FAD), melanin, keratin, and so on) and exogenous fluorophores, being exploited in fluorescence lifetime imaging for a plethora of experimental approaches [3]. Extrinsic contrast agents, such as organic fluorophores and genetically expressed fluorescent proteins, are suitable for exploring the skin’s structure and biochemistry with multiphoton microscopy, whereas endogenous contrast agents help investigate the main components of the skin [4]. Flavin proteins and NAD(P)H are mainly located in mitochondria, but NAD(P)H may also be present in the cytosol. Nonetheless, the fluorescence resulting from reduced pyridine nucleotides and oxidized flavin proteins is very useful in portraying the keratinocytes [4]. Besides, the endogenous fluorescence of keratin can be exploited to emphasize the presence of this structural protein in the stratum corneum, where it is abundant [5]. Based on their fluorescence, collagen and elastin are also observable in the dermis [6]. However, some isoforms of collagen, such as type I and II, are not visible in TPEF due to their non-centrosymmetric molecular structure, being observable only in SHG microscopy [7]. Other applications of TPEF include the non-invasive assessment of size variation of cell nuclei, blood vessel architecture, and inflammatory states, which may serve as valuable indicators of pathological transformation in tissues [2].

In parallel, SHG microscopy exploits the interaction of two incident photons with a non-centrosymmetric molecule, resulting in a single emitted photon with halved energy via a nonlinear process involving virtual states [7]. SHG imaging is, therefore, suitable for the non-invasive assessment of non-centrosymmetric structures, such as collagen fibers or myosin filaments in various biological specimens. Collagen imaging is particularly important in the skin, as it is the main structural protein in the extracellular matrix and the modifications of the collagen-rich matrix often correlate with neoplastic transformation in humans and living organisms [8].

TPEF has shown promising results in exploring the morphology of skin cancers, and several MPM features proved to be highly relevant to differentiate these pathologies from the surrounding healthy tissues [9]. A commercial multiphoton tomograph for clinical applications also became available recently, indicating the efficiency of in vivo dermal SHG and TPEF tomography. However, even though the specificity of NMSC detection is remarkable, the ex vivo discrimination between different cancer types needs some refinement. Accordingly, obtaining some additional spectroscopic information will help significantly increase the accuracy of this microscopy technique in clinical practice [9].

One of the pioneering studies in the field conducted by Cicchi et al. on a tumor biopsy taken from a patient with BCC highlighted an increase in fluorescence intensity in cancerous tissue compared to the healthy samples [10]. Finally, the use of aminolevulinic acid as a contrast agent has been demonstrated to increase the contrast in tumor border detection, supporting the use of MPM for the in vivo non-invasive imaging of BCC [10]. One year later, Paoli et al. used MPM to assess the morphologic features of NMSC on freshly excised specimens from 14 patients [11]. The microscopy was carried out ex vivo using a femtosecond pulsed laser at 780 nm and an ×40/0.8 objective, and the autofluorescence was detected in the range of 450–530 nm. Conventional histopathological criteria such as bowenoid dysplasia, multinucleated cells, hyperkeratosis in SCC, and peripheral palisading of tumor cells in superficial BCC were distinguished with MPM. However, characteristic tumor aggregates were found in only one of the three investigated nodular BCC due to limited imaging depth, highlighting that MPM could be potentially applied for non-invasive diagnostics of BCC and SCC, whereas the ability to characterize BCC requires further investigation [11]. In parallel, Ericson and colleagues revealed that cell nuclei distribution might be an important parameter that can be used for discriminating between tumor and normal tissue ex vivo [12]. A recent study also highlighted that key histological characteristics present in conventional histology might also be detected with two-photon fluorescence. In addition, it was reported that the diagnosis of BCC displayed perfect accuracy (100% sensitivity and specificity), while the diagnosis of SCC reached high accuracy (89% sensitivity and 100% specificity) with TPEF [13]. Finally, several reports also describe the utility of MPM in imaging the dermo-epidermal junction (DEJ) in unstained fixed tissues, providing valuable cues for histopathologists to identify the onset of NMSC in patients [2].

TPEF was also used for exploring melanoma tumors and their precursors in clinical cohorts. In one study on imaging melanoma in nevi, highly fluorescence clusters were found, indicating pigmented cells mainly in the basal layer of the epidermis [14]. The autofluorescence of the different tissue structures was detected with 760 nm (NAD(P)H, elastin, pigmented cells) and 840 nm (pigmented cells, collagen, SHG) femtosecond laser pulses. In MPM microscopy, pigmented cells showed a much brighter fluorescence than the surrounding non-pigmented or less pigmented corneocytes [14]. A clinical trial on 250 patients also confirmed that melanoma cells fluoresce much more brightly than surrounding cells in MPM, and morphological differences could also be seen in pathological tissues versus normal ones [15]. In addition, in a clinical cohort of 115 patients, MPM showed a sensitivity of 95% and a specificity of 97% for the diagnosis of melanoma tumors [16]. Logistic regression analysis was performed to identify the most significant diagnostic criteria. It was found that architectural disarray of the epidermis, poorly defined keratinocyte cell borders, and the presence of pleomorphic and dendritic cells, were the most important hallmarks of melanoma [16]. In parallel, a recent study highlighted that MPM features such as cell symmetry, cell distance, cell density, cell and nucleus contrast, nucleus-cell ratio, and homogeneity of cytoplasm might be very effective in differentiating between malignant melanomas, lesions, and healthy skin [17]. Despite these optimistic results, fluorescence intensity does not highlight tumor cells with sufficient specificity to cause this technique to be applicable in clinical practice; therefore, more extensive studies with MPM are needed for future clinical application.

Although MPM proved comparable efficiency with RCM in (skin) cancer screening, many precautions must still be taken into account regarding the use of this technique in clinical cohorts. One of the most important things to consider is the skin’s stratified structure, with different structural layers having very different refractive indices. Briefly, in multiphoton imaging, this multi-layered structure with varying indices of refraction causes spherical aberration and distorts the excitation focus, resulting in signal loss and a reduction in image resolution [4]. It influences the imaging depth as well. Multiphoton imaging presents an imaging depth of 150 to 200 μm in the skin and 2 mm in the brain. The difference in penetration depth between the brain and the skin may be explained by the fact that deep brain imaging targets relatively larger vascular structures. In contrast, skin imaging focuses more on detailed features such as the morphology of individual cells or elastin fibers [4]. Additionally, skin imaging is based on endogenous fluorophores such as NAD(P)H, which emit at approximately 450 nm, a wavelength that has a short mean free path in tissues, while imaging deep in the brain is often based on exogenous fluorophores emitting more at red wavelengths [4]. Another significant drawback of multiphoton skin imaging is represented by photo-damage. Two or higher photon excitation of endogenous and exogenous fluorophores results in oxidative photo-damage, triggering a photo-damage pathway similar to that of ultraviolet irradiation in tissues [18]. The photo-activation of these fluorophores is accompanied by an abundant production of reactive oxygen species (ROS), which consecutively triggers the biochemical damage of cells [19]. Flavin-containing oxidases have been indicated as one of the primary endogenous targets for photo-damage in cells [20].

Nonetheless, although there are still some technological and financial limitations that need to be overcome to improve MPM performances, the results obtained in clinical studies with this technique so far suggest that it may soon become an important tool in the routine care of (skin) cancer patients, which may help in guiding the diagnosis and clinical decisions in this hard-to-treat disease.

### 2.3. Optical Coherence Tomography (OCT)

OCT is another non-invasive imaging technique frequently exploited for in vivo skin analysis to obtain real-time cross-sectional and en-face images of the tissue. OCT combines near-infrared and infrared radiation (800–1300 nm), enabling the detection of the back-scattered light waves at indexes of refraction mismatched in the skin’s internal structure [16]. The back-scattered light from the target area is recombined with that from a reference mirror; yet, when both path lengths are matched within the so-called coherence length of the light source, an interference occurs, which may be of micrometric size [85]. Therefore, it is possible to detect signals from different depths and determine from which depth in the skin the signal originates. The radiation’s wavelength determines the penetration depth. Therefore, an OCT device generally provides a scan depth of 2 mm, lateral resolution of 3–8 µm, and vertical resolution of 5–10 µm [16,86]. Although the OCT resolution is inferior to RCM, OCT proved effective in exploring more profound depths than RCM and capturing structural changes in the targeted tumor areas [87]. Besides, OCT comes with the advantage of providing real-time, dynamic images of the skin in a very short time, usually less than a minute. It also has a large field of view, so this technique can thoroughly cover an area of 6 × 6 mm^2^ [88].

OCT can generate both two-dimensional (2D) and three-dimensional (3D) images suitable for the depiction of thin tissue layers. Two-dimensional data (b-scans) are obtained by moving the beam across the skin to acquire vertical, cross-sectional scans; yet, 3D data are obtained by translating the beam in two directions over a surface area. Moreover, the en face, also known as the horizontal mode, which provides images similar to those in dermoscopy or RCM can be also found incorporated within most available clinical instruments [76]. There are several different types of OCT that have been developed for specific purposes, such as Fourier domain-OCT (FD-OCT) and high-definition OCT (HD-OCT). FD-OCT, which detects the spectral components of the light source either with a spectrometer or a wavelength-swept laser, offers shorter acquisition times and higher sensitivity, becoming for a long time now the standard in dermatological research [89]. HD-OCT even comes with an enhanced resolution than the FD-OCT, being able to provide details of cellular structures with accuracy [90]. Other OCT systems, such as angiographic OCT or dynamic OCT, have been developed to detect the blood flow and capture high-resolution images of the skin microvasculature [91,92]. Hence, recent advances in the field culminated in the development of line-field confocal optical coherence tomography (LC-OCT), which combines the principles of both OCT and RCM, allowing imaging at a greater resolution than other types of OCT and with a higher detection depth compared to RCM [93].

During the last decade, OCT has evolved from being an interesting scientific tool in the laboratory to a helpful bedside companion diagnostic device for clinical diagnosis and monitoring of skin cancer patients. The most significant applications of OCT in dermatology have thus far been in the diagnosing, delineating, and treatment monitoring of NMSC, especially BCCs [86]. In particular for BCC, OCT allows for the visualization of architectural changes down to the middle dermis but without cellular resolution [93]. OCT can also be used to determine the tumor thickness and the subtype of BCC [94,95], but also to evaluate the therapeutic response following non-surgical therapies [96]. Yet, pigmented lesions continue to pose significant challenges in OCT imaging and, in diagnosing CM, OCT is not as accurate as dermoscopy or RCM. Additionally, OCT presents a limited tissue penetration depth, which is of particular concern regarding the evaluation and delineation of non-superficial malignant skin tumors [86]. Here, we will briefly describe the current status of OCT in the most important cutaneous cancer subtypes.

BCC is the most studied cutaneous tumor by OCT. The results of a multicenter prospective study highlighted that the sensitivity and specificity of OCT for the diagnosis of BCC were 95.7% and 75.3%, respectively. Moreover, the use of OCT in combination with the clinical examination increased the diagnostic accuracy to 87.4% from 65.8% than it was for clinical examination alone [88]. In line with this observation, another multicenter prospective study confirmed that OCT has significantly higher sensitivity and specificity for the diagnosis of BCC when compared with dermoscopic evaluation. Overall, the sensitivity and specificity of OCT were 92% and 80%, respectively, whereas the sensitivity and specificity of dermoscopy were reported to be at 78.6% and 55.6%, respectively. Notably, the authors found that they could reduce the number of biopsies in their cohort by 35% if patients benefit from additional OCT evaluation [97]. Besides this diagnostic role, OCT may also be used to evaluate tumor depth and margin, assisting physicians to deliver more personalized treatments to the affected patients [98]. The tumor depths of BCC measured using OCT were similar to those measured during histological examinations [98]. Moreover, several other studies highlighted that OCT might be a valuable tool to assess the treatment response after various treatment modalities such as PDT [99], topical [96], and systemic therapies [100]. Finally, the evaluation of abnormal BCC vasculature with angiographic OCT is an area of active investigation at the moment [101]. Given that the vascular morphology correlates with the subtype of BCC, the angiographic and dynamic OCT proved to be an important tool to differentiate the common BCC subtypes and guide the therapeutic decisions in BCC [102].

SCC tumorigenesis covers a plethora of pathological entities, ranging from local patches of dysplastic cells to AK and full-blown SCC tumors, all with increased histological similarities. All of these tumors are usually hyperkeratotic and erythematous to varying degrees, posing significant diagnostic challenges and calling for numerous biopsies [86]. The current literature highlights that invasive SCC, in situ SCC, and AK with thick hyperkeratotic scales may be difficult to explore with structural OCT images mainly because the hyperkeratotic epidermis of these lesions prevents OCT from obtaining insights into deeper skin layers and hence observing the dermal–epidermal junction [101]. Although a meta-analysis reported high pooled sensitivity (92.3%) and specificity (99.5%) of OCT for the diagnosis of SCC and highlighted its ability to distinguish between SCC and BCC, at the moment, more elaborated studies need to be conducted to validate the utility of OCT in the clinical diagnosis of SCC [103]. The development and introduction of HD-OCT and angiographic OCT, which proved to be highly effective in distinguishing between AK and SCC with increased accuracy, hold promise for revolutionizing SCC diagnostic approaches in clinical settings [104,105]. Nonetheless, there are also several reports that emphasize a role for OCT in monitoring AK after treatment with various modalities, such as ingenol mebutate [106] and cryosurgery [107].

The restricted imaging depth of OCT and the lack of parameters useful for diagnosis, especially for the differentiation between (dysplastic) nevi and melanomas are the main factors that have limited the use of OCT in CM [86]. There are several studies that suggest that novel OCT devices may have a potential role in the diagnosis of CM. In a multicentre pilot study examining 93 melanocytic skin lesions, Gambichler et al. showed that that the sensitivity and specificity of HD-OCT for the diagnosis of CM were around 74% and 92.4%, respectively. However, HD-OCT indicated high false negative rates in very thin melanomas and high false positive rates in dysplastic naevi, suggesting the diagnostic performance of HD-OCT of MSL should be reassessed in other clinical settings [108]. Moreover, the fact that the vessel patterns may be associated with the CM tumor stage prompted other researchers to harness the potential of angiographic OCT for CM diagnosis [91]. Finally, other authors have focused on the potential of LC-OCT in discriminating between nevi and melanomas [109]. Schuh et al. examined 84 melanocytic lesions with LC-OCT and 36 other melanocytic lesions with RCM. OCT had a 93% sensitivity and 100% specificity compared to RCM (93% sensitivity, 95% specificity) for diagnosing a melanoma (vs. all types of nevi) [110]. Both devices falsely diagnosed dysplastic nevi as non-dysplastic (43% sensitivity for dysplastic nevus diagnosis) and hence showed no differences in performance when compared. The most significant criteria for diagnosing a melanoma with LC-OCT were irregular honeycombed patterns (92% occurrence rate), the presence of pagetoid spread (89% occurrence rate), and the absence of dermal nests (23% occurrence rate) [110].

LC-OCT is gaining considerable interest in (dermato-) oncology, as the results achieved by combining RCM and OCT are encouraging by compensating the limitations of each device. Of particular importance, there are some reports highlighting that LC-OCT may guide the detection of the most relevant diagnostic or prognostic areas in vivo, allowing for accurate and targeted biopsies and hence precise downstream histopathology and molecular profiling in cancers [111]. OCT and RCM-assisted sampling might also play a pivotal role in monitoring therapeutic responses at a cellular level by tracking the tumor mutational burden or evaluating the expression of immune biomarkers such as PD-L1 in cutaneous malignancies [111].

### 2.4. High-Frequency Ultrasound (HFUS)

Ultrasound emerged in the practice of dermatology about 50 years ago when two independent groups used the technique to measure skin thickness in normal and pathological settings. Since then, the method has gained popularity in several European and South American countries, culminating in its incorporation in resident training and indexing for complete reimbursement in healthcare systems in many countries. Ultrasound enables high-resolution imaging of the skin from the stratum corneum down to the deep fascia [112]. Until the last decade, the high frequency sonography (20 MHz) and mid-frequency sonography (7.5–15 MHz) were the mainstay approaches in the diagnosis of cutaneous tumors [17]. Nonetheless, novel ultrasound technologies, including high-frequency ultrasound (HFUS, 20–30 MHz) and ultra-high-frequency ultrasound (UHFUS, >30 MHz), have further enhanced imaging clarity and revolutionized the applications of ultrasound for clinical use in dermatology [112]. The frequency of HFUS is proportional to the resolution of the image and inversely proportional to the penetration depth of sound waves. For instance, HFUS with a frequency of 20 MHz can provide a resolution of 50–300 µm and may achieve a penetrating depth of 6–7 mm. In contrast, HFUS with a frequency of 50 MHz can provide a higher resolution of 39–120 µm and a lower penetrating depth of 4 mm [16]. HFUS remains a versatile, painless, noninvasive procedure that can be performed virtually everywhere and can be readily repeated.

Conventional ultrasound devices use a transducer and piezoelectric crystals to create images of the internal body structures when stimulated by an electrical voltage and send them to a computer to generate sonograms [112]. The basic modes of ultrasound include the Doppler method, A-mode, and B-mode [17]. A lot of devices may incorporate color Doppler analysis for the characterization of blood flow and vessel morphology within the suspected skin lesions [112]. Regarding the amplitude mode (A-mode), the computer generates a one-dimensional line graph that can be used to interpret echogenicity at various distances from the probe [112,113]. The so-called echogenicity represents the internal echo pattern—or the ability of a structure to reflect sound waves relative to its surrounding tissue [114]. In addition, in brightness mode (B-mode), an image is generated with different intensities of brightness. That is, the brightness of each pixel correlates with the amplitude of the echo. The images with high intensity echoes are termed echogenic or hyperechoic; in contrast, those with low intensity are called hypoechoic; hence, the ones without echoes are named anechoic or echolucent [28]. The echogenicity of each tissue is determined by the speed at which the sound wave can pass through it and the quantity and intensity of echoes captured by the device. In normal skin, the echogenicity of each layer is greatly influenced by its main component such as keratin (in the epidermis), collagen (in the dermis), or fat lobules (in subcutaneous tissue). Therefore, the dermis appears highly echogenic and the epidermis is presented as a hypoechoic line-sharply demarcated by other skin structures, and the subcutaneous tissue, as a hypoechoic layer [28]. Remarkably, in HFUS, cutaneous tumors, either malignant or benign, present as echogenic structures in contrast to the neighboring healthy tissue. Nonetheless, HFUS offers the possibility of assessing the lesion shape, performing longitudinal, axial, and transverse measurements, to identify the tumor contour and invasion of adjacent structures (bone, cartilage, or muscle) [28].

Given that during BCC development the lower density (hypoechoic) tumor masses are replacing (more hyperechoic) collagen within the tissue, ultrasonography is currently regarded as a promising tool that may help diagnose BCC. Notably, the existence of this hypoechoic mass harboring hyperechogenic areas in the interior due to corneous cysts or nests of apoptotic cells (termed “flower cotton”) helped differentiate BCC from SCC and CM with increased accuracy [115]. HFUS also proved effective in estimating the BCCs size (thickness and diameter), delineating tumor margins, and helping in Mohs surgical planning [116]. For instance, a study conducted by de Oliveira Barcaui on 83 lesions from 67 patients with clinical and dermoscopic diagnosis of BCC highlighted that HFUS (22-MHz) is able to localize deep tumor margins in BCC, showing similar results to histopathological measurements. HFUS had a sensitivity of 96%, specificity of 84%, and accuracy of 91% for measurement of deep tumor margins [117]. In parallel, Khlebnikova et al. showed that in cases of BCCs with thicknesses of ≤1 mm, there was a high correlation (r = 0.870) of the tumor spread depth between micromorphological measurements and the results obtained using a 75 MHz transducer. Conversely, in cases of BCCs with thicknesses of >1 mm, a positive correlation (r = 0.951) of the tumor spread depth was reported between the histomorphometry and 30 MHz transducer measurements [118]. Other authors have also employed HFUS for differentiating between invasive BCC and non-invasive BCCs. A suggestive study in this setting is that conducted by Wang et al. on one hundred patients diagnosed with BCCs. Approximately 60.5% of invasive BCCs presented with an irregular growth pattern, whereas 89.5% of non-invasive BCCs displayed a nodular or crawling pattern (*p* < 0.001) [119]. Regarding the intralesional hyperechoic spot distribution, which is a main feature of BCC in HFUS, invasive and non-invasive BCCs tended to be clustered and absent/scattered, respectively (55.8% vs. 91.2%) (*p* < 0.001). The presence of seven or more hyperechogenic spots within the lesion has been associated with highly invasive BCC histological subtypes, such as micronodular, infiltrative, or morpheaform and metatypical forms. On the basis of the aforementioned features, a prediction model was established with accuracies of 84.0% and 76.7%, respectively, in the pilot and validation cohorts, suggesting that HFUS may be a valuable tool for the differentiation of the BCCs subtypes in the clinical management of this disease [119]. In addition, HFUS may be also used to evaluate the vascularization and blood flow of BCC tumors [115] as well as in monitoring the therapeutic responses to high-dose ionizing radiation therapy [46] and Hedgehog inhibitors [120] in a non-invasive manner.

The visual presentations of SCC are quite similar to those of some high-risk BCC forms which can lead to misdiagnosis in certain cases [121]. SCC appears as a hypoechogenic mass in relation to the surrounding tissue, without clear specific features that allow the differentiation from other NMSC or skin lesions. Compared with BCC, cSCC has a more aggressive and unpredictable behavior, being more likely to invade soft tissues, cartilage and adjacent bone, leading to poor survival outcomes in the affected patients. Therefore, it is an urge to find more reliable tools that may help in distinguishing between BCC and SCC for optimal treatment and improved clinical outcomes [122]. In a pioneering study in the field that included 4338 skin lesions, Wortsman et al. showed that the addition of ultrasound increased the correctness of diagnosis from 73% to 97% [123]. Nonetheless, Tong Chen et al. have recently shown that the internal hyperechoic spots, alongside several other HFUS features, such as lesion size, thickness, posterior acoustic shadowing, and Doppler vascularity pattern, are useful for differential diagnoses between high-risk BCC and SCC. Based on the aforementioned five features, an optimal discrimination model was established with a sensitivity of 91.2%, a specificity of 87.7%, and an accuracy of 89.5% [121]. It is also worth mentioning that HFUS seems an ideal tool for determining the local aggressiveness of a tumor [122] and guiding the therapeutic approaches [124]. HFUS may provide valuable information regarding the tumor size and depth of invasion, which is of great importance when planning the extent of surgery in the clinical management of SCC [122]. Nonetheless, due to the inflammation and the hyperkeratotic characteristic of some SCC, HFUS may have a decreased accuracy in investigating the size of SCCs or assessing its depth of invasion [122,125]. The overall low resolution, lack of functional contrast, and image quality are other limitations of HFUS. For this reason, HFUS should be implemented with care in the clinical setting of SCC and always coupled with a validated diagnostic technique such as dermoscopy [16].

Ultrasound is emerging as a promising approach for the evaluation, staging, and follow-up of patients with CM as it is a non-invasive and inexpensive imaging method and it shows more sensitivity and specificity than physical examinations [126]. In HFUS, melanocytic lesions appear as a hypoechoic, homogeneous mass that is oval- or spindle-shaped. Benign lesions are also characterized by irregular echogenicity, which means that they are hard to differentiate using HFUS, leading to the overestimation of tumor size in cases with nevus–melanoma association. However, the examination of the tumoral vasculature with color Doppler was shown to enhance the diagnostic accuracy of melanocytic lesions, as malignant entities present a more intense vascularization network than benign nevi, predominantly composed of arterial vessels of low flow [28]. Two decades ago, Bessoud et al. showed that in 111 patients with pigmented skin lesions, HFUS alone provides 100% sensitivity and 100% specificity in the distinction of melanoma/nevi from other skin lesions and 100% sensitivity and 32% specificity in the distinction of melanomas from nonmelanoma lesions [127]. At the same time, color Doppler detection of intralesional vessels had a 100% specificity and 34% sensitivity in the distinction of melanomas from other pigmented skin lesions. Interestingly, for all the samples analyzed, the sonographic and histologic measurement of melanoma thickness showed a strong correlation [127]. Similar degrees of correspondence between tumor depths measured using HFUS and histopathology were reported by Meyer and his team [128]. US also proved useful in detecting the metastases that are localized deeper in soft tissues and are impalpable in clinical examinations [129]. In the ultrasound image, the metastatic lesions localize in the hypodermis and have an irregular shape in contrast to the nevic lesions, which present discoid shape and are located in the dermis. Metastatic lymph nodes may also be seen in ultrasound imaging as round, well-defined structures with a hypoechoic or echolucent center [28]. With regard to lymph node assessments with US, this tool shows an overall sensitivity of only 24% for the detection of metastases in SLNs mapped on pre-operative lymphoscintigraphy. This low rate is due to the inability to detect micrometastases. Nonetheless, it was further observed that the sensitivity of the assay increased when the tumor size exceeded 4.5 mm in diameter. Therefore, pre-surgical US cannot replace SNLB in the evaluation of regional lymph nodes [130]. Of particular importance, Solivetti et al. recommend the use of a transducer of a different frequency to detect metastatic lesions because, depending on the site and size of a lesion, it can be visible in a particular frequency but completely invisible in another [131].

### 2.5. Terahertz Pulsed Imaging (TPI)

Due to the non-ionizing and non-invasive nature of THz radiation, alongside its high sensitivity to water, THz imaging emerged as a highly attractive tool for biomedical applications within recent years [132]. Terahertz (THz) radiation—also termed as submillimeter radiation, THz waves, THz light, or T-rays—lies between the microwave and infrared regions of the electromagnetic (EM) spectrum (0.1–10 THz) and corresponds to wavelengths from 0.03 mm to 3 mm [133]. Given that the energy in this frequency range excites the intermolecular interactions, such as the rotational and vibrational modes of molecules, the data acquired can provide important spectroscopic information on the material under study. That is, both the amplitude and phase of the terahertz pulse can be obtained, from which the absorption coefficient and refractive index of a medium can be properly determined [29]. Because the absorption is greatly influenced by the chemical constituents of the medium, the specific signatures of a disease can be measured to offer diagnostic information. In addition, the refractive index of the structure being imaged may also provide morphological and functional information that may be relevant either for pathologists and dermatologists in clinical setting [134].

The electromagnetic waves constituting THz radiation can be generated by two types of systems: (1) the continuous wave (CW) systems, which produce a single or several discrete frequencies and (2) pulsed systems, characterized by a broadband frequency output, dating back to 1995 when they had been proposed for the first time by Hu and Nuss as potential imaging tools. All these systems display significant differences in terms of optical properties, frequencies, and sensitivity. THz is currently exploited for both in vivo and ex vivo clinical applications in the analysis of epithelial cancers [134]. However, water represents an important issue in THz imaging; that is, since it is particularly sensitive to water content, THz radiation exhibits a strong absorption that in turn limits its penetration depth in fresh tissues [135]. Therefore, THz waves can penetrate only a few hundred microns into the human skin [136]. Several groups have proposed freezing techniques to increase the penetration depth of THz radiation into wet tissues, as THz absorption by water molecules decreases dramatically in ice [137]. THz also showed limitations when performing ex vivo measurements on resected tissues, usually linked to saline uptake from the sample storage environment, changes in hydration level during the measurement, as well as other temperature- and humidity-dependent interfering effects [134]. However, the high sensitivity of THz waves to water content in the tissues remains the key contrast factor in biomedical applications; it allows distinguishing between various types of tissues as an endogenous marker, allowing a clear demarcation of healthy and pathological tissues [134]. In particular for cancers, the increased blood supply to the affected tissues is the main factor that leads to an increase in tissue water content, supporting the use of water contrast in THz cancer imaging [138].

BCC is the best studied cutaneous tumor with THz imaging. Twenty years ago, Woodward et al. proved that TPI in reflection geometry may be an ideal tool for assessing skin tissue and related cancers both in vivo and in vitro [139]. Hence, they showed that the sensitivity of THz radiation to polar molecules such as water may be effective in determining the lateral spread of BCCs pre-operatively, highlighting that this macroscopic technique may, in the future, help plan surgery. Resolutions at 1 THz of 350 μm laterally and 40 μm axially in skin were reached with this system. Notably, BCC has shown a positive THz contrast and inflammation, whereas scar tissue displayed a negative THz contrast compared to normal tissue [139]. Soon after that, Wallace et al. also reported changes in THz properties in malignant tissue in comparison with normal tissue, which correlated well with regions of BCC seen in histology [140]. The lateral and axial resolutions attainable with this system at 3 THz were of about 150 μm and 20 μm, respectively, similar to other imaging techniques routinely used in medicine, suggesting that THz is a viable imaging modality for skin cancer biomedical applications [140]. Moreover, there are also several reports that highlight that THz imaging may be used to investigate the mechanisms by which the skin responds to various treatments, mainly those that are topically applied on the skin. A suggestive study in this context is that conducted by Ramos-Soto et al., which assessed the effects of common moisturizer ingredients on an excised sample of porcine skin [141]. Notably, this may lead to the development of similar in vivo THz approaches to assess the therapeutic effects of the topical application of 5-fluorouracil or imiquimod in BCC patients.

Studies on SCC with THz are scarce. There is just a single report by Srivastava et al. that describes a novel 3D reconstructive imaging system utilizing THz radiation that may be used for non-invasive skin cancer diagnosis [142]. The reconstructed images of BCC and SCC were characterized by increased cellularity and disorganization as features that can differentiate between healthy and diseased skin. To investigate the differences between the NMSCs and establish an accurate diagnosis, the authors recommended corroborating the images of BCC and SCC skin specimens with histopathology information and THz images [142].

THz imaging has also been successfully applied in CM. In a study published in 2016, Zaytsev et al. reported significant contrast in THz dielectric permittivity responses of healthy skin and dysplastic and non-dysplastic skin nevi, which may be effective in detecting the precursor of melanoma, proving the efficiency of THz-pulsed radiation in the early diagnosis of CM [143]. There are also several reports that describe THz pulse imaging of CMs dehydrated by ethanol and embedded in paraffin. For instance, by combining THz imaging (0.2–1.4 THz) with multi-structural element mathematical morphology, Li et al. developed an approach that may facilitate a relatively accurate determination of the areas of normal and cancerous tissues [144]. They focused mainly on comparing the refractive index and the absorption coefficient of CMs with that of adjacent healthy skin tissues. The refractive index of malignant tissues varied between 0.2 and 0.7 THz, while that of normal and fat tissues remain almost unchanged. Additionally, the absorption of CM specimens was higher, with a maximum at 0.37 THz. Therefore, the authors suggested that their method of THz pulse imaging combined with mathematical morphology may be effective for the safe surgical removal of CM tumors [144]. Although the diagnostic accuracy of THz imaging is quite satisfactory, it does not have the ability to discriminate between skin cancer subtypes. Although TPI may be able to assist surgical decision/planning in the future, there is still a long way to go before its approval for use in clinical practice [17].

Nonetheless, several reports highlighted the potential of THz radiation in detecting certain epigenetic alterations, such as DNA methylation, in cancers [145,146]. DNA methylation plays a vital role in regulating gene expression in physiological settings; hence, increased methylation in the promoter region of tumor suppressor genes (TSGs) correlates with their suppression and is a hallmark of cancer [147]. Notably, a THz resonance fingerprint of methylated malignant DNA has been reported at 1.65 THz for several types of cancer. Given that epigenetic alterations occur prior to genetic mutations, THz radiation may serve as a valuable tool that may help in the early detection of skin tumors [148]. Moreover, the current literature highlights that this epigenetic modification could be manipulated to the state of demethylation using a high-power THz radiation [30]. Demethylation enhances the efficiency of conventional cancer treatments, facilitating transcriptional reprogramming, cell apoptosis, and tumor shrinkage [149]. The hypermethylation status of TSGs may be reversed with the use of DNA methyltransferase inhibitors (DNMTis)-cytosine analogue inhibitors and non-nucleotide analogue inhibitors. However, DNMTis showed limited efficiency and multiple severe adverse effects, including hematological toxicity, when used in the treatment of solid cancers. Therefore, because THz radiation is a non-invasive and non-ionizing technique using electromagnetic waves of a specific resonance frequency, it can be a good solution to design a novel cancer therapy with less side effects than the traditional ones [30].

### 2.6. Magnetic Resonance Imaging (MRI)

The first attempts to carry out the non-invasive MRI mapping of the skin were conducted by Bittoun et al. [150] and Richard et al. [151,152] and date back to the early 1990s. Today, after more than two decades of continuous employment of MRI in clinical practice, the utility of this diagnostic imaging modality has expanded beyond expectations to become an essential tool for cancer diagnosis and management worldwide. For skin cancer, the applications of MRI imaging include the evaluation of tumor extension, nodal involvement, and distant metastasis. Moreover, MRI may be useful in assessing intratumoral characteristics of cutaneous tumors, such as homogeneity, cyst formation, and hemorrage, serving as a potential companion technique for determining the histological subtypes of skin tumors [153].

MRI relies on nuclear magnetic resonance (NMR), which exploits the magnetic properties of protons or other nuclei in an external magnetic field upon exposure to radiofrequency (RF) electromagnetic waves of a specific resonance frequency [154]. Several types of pulse sequences have been defined to highlight differences in the signals of various soft tissues. The most common and most basic of pulse sequences include T1-weighted and T2-weighted sequences. Among the entities associated with a high signal intensity on T1-weighted images should be mentioned fat, proteins, blood (methemoglobin), some forms of calcium, melanin, and gadolinium (a contrast agent). In contrast, tissues that show a high signal on T2-weighted images are characterized by fluid-containing structures (e.g., cysts and cerebrospinal fluid) and hence pathologic states causing increased extracellular fluid and water content (e.g., inflammatory and neoplastic diseases) [155]. Therefore, MRI enables the observation of anatomic structures, physiological functions, and molecular architecture of tissues in a non-invasive manner.

In most clinical applications, MRI is employed to differentiate the different forms of skin cancers and help in the localization and delineation of those tumors that may be difficult to assess because of their topography [156,157]. MRI also allows determining the degree of invasion of malignant tumors within deeper soft tissues and hence can measure their size and thickness [156,158,159]. For this reason, the National Comprehensive Cancer Network guidelines recommend that the local extension of high-risk SCC and BCC (of any size on the face, hands, and feet and larger than 2 cm on the trunk or extremities) to be evaluated by radiological approaches [160,161]. Because of its higher sensitivity and superior soft tissue contrast, MRI is preferred over computed tomography (CT) if perineural disease or deep soft tissue involvement is suspected. MRI is a useful tool for assessing muscle tendon or bone marrow invasion; in contrast, CT can reveal cortical bone invasion [153]. Given the good diagnostic performances achieved in the detection of metastases, the Dermatologic Cooperative Oncology Group and the updated Swiss Guidelines suggested the use of whole body-MRI as an alternative to 18-flurodeoxyglucose (FDG) PET/CT for the staging of high-risk and metastatic (stage III or IV) melanoma and for the follow-up of stage IIC or higher CM patients [162]. Nonetheless, although more expensive than CT, MRI provides many advantages, including a higher field of view, spatial resolution under 100 μm and an excellent contrast, all of them without utilizing harmful ionizing radiations [162,163].

MRI findings on BCCs on the head and neck regions revealed the nose as the most common anatomical site of their occurrence. The examined nodular and infiltrating BCC exhibited well-demarcated deep tumor margin without peritumoral fat stranding or protruding into subcutaneous tissue, with an estimated maximum diameter of 12.7 mm. Furthermore, the authors suggested the intratumoral T2-hyperintense foci as a main feature of BCC tumors, as it seldom appeared on the examined SCC tumors in their study. However, in all eight BCCs with intratumoral T2-hyperintense foci, a radiologic–pathologic correlation on a one-to-one basis was performed and revealed cystic cavities filled with mucinous contents within BCCs areas corresponding to the intratumoral T2-hyperintense foci, suggesting that this imaging finding may be very helpful in the distinction of BCCs from SCCs [156].

SCCs rarely occur on the nose (5–14%), being more frequently reported on the lateral areas of the face, including the auricle and auricular, buccal, parotid-masseteric, and temporal areas. On MRI, SCC of the head and neck exhibited a flattened configuration with well-demarcated deep tumor margins and hence more aggressive features than BCC, such as superficial ulcer formation, protrusion into the subcutaneous tissue, and peritumoral fat stranding. The maximum diameter of SCC in MRI was evaluated to be around 23.5 mm. Nonetheless, the SCC in this study frequently displayed heterogeneous signal intensities on T2-weighted images, which may be a hallmark of cystic necrosis and aggressive behavior in these tumors [156]. Previous reports also highlighted that a considerable percentage of SCC have a non-specific pattern of hypointensity on T1-weighted images as well as iso- and hyperintensity on T2-weighted images [164,165]. Nonetheless, the T2-heterogeneous signal intensities, including non-enhanced regions, on contrast-enhanced images, were linked with a fulminant SCC progression and cystic necrosis in these tumors [164].

In CM, the presence of blood products and melanin may be harnessed by MRI to generate distinctive features that may help in detecting and delineating melanocytic lesions [166]. Notably, approximately half of the melanoma metastases appear hyperintense on T1-weighted images relative to the dermis probably due to melanin deposition or intratumoral hemorrhage, whereas the remaining half of CM metastases shows isointensity relative to the dermis [153]. Cerebral melanoma metastases also appear as hyperintense on T1-weighted images and hypointense on T2-weighted images and, in this case, hemorrhage in the lesion seems to have a greater influence on this unique appearance than does melanin [167]. A relative recent study led by Kawaguchi et al. on a total of 16 patients with primary CM and 49 with primary SCC confirmed that MRI features may help discriminating between primary CM and primary SCC with increased accuracy [168]. According to their study, intratumoral T1 hyperintensity relative to that of the dermis was more frequently reported in CM than in SCC (50% vs. 4 %, *p* < 0.01). In contrast, superficial depression, superficial irregular margins, and reticular or linear T2 hyperintensity were more frequently observed in SCC than in CM. No significant differences in terms of ill-defined deep tumor margin, peritumoral fat stranding, or T2 heterogeneity were noted between CM and SCC. Nonetheless, in this study, CM occurred predominantly on the foot. The depth of invasion in CM was intradermal in 13%, subcutaneous tissue in 81%, and bone in 6% of cases, whereas that in SCC was subcutaneous tissue in 84% and bone in 6% of cases [168]. It also should be noted that performing MRI scans in patients with metastatic melanoma aid in the early detection of brain metastases before neurological symptoms occur. In one study conducted on 116 melanoma patients, one third of the brain metastases that developed within the first 2 years after the diagnosis of metastatic melanoma were detected asymptomatically by six monthly screening MRI scans [169]. Moreover, the same study reported that the three monthly follow-up MRI scans led to treatment strategy changes in more than 45% of patients. Changes in the treatment strategies after follow-up MRI scans occurred less frequently in patients with durable responses to immune checkpoint inhibitors (ICI) [169]. In line with these observations, a recent single center study on 2745 melanoma patients analyzed between 2011 and 2017 reported that while the overall incidence of brain metastases remains high, there is a decreasing incidence of brain metastases over the follow-up period [170]. This trend may be attributed to the early therapeutic interventions enabled by MRI scans in patients with asymptomatic progressive disease. In addition, another study reported that the early detection of asymptomatic intracranial progression using follow-up brain imaging was cost effective due to less use of neurosurgical interventions and lower expenses for the management of neurological symptoms [171]. However, according to the aforementioned studies, the most important determinants that are relevant for the optimal scan interval are the melanoma location and type, metastatic sites, mutational status, and LDH-level at diagnosis [169]. Regarding the treatment changes performed after follow-up MRI scans, the most common ones consisted of switching from BRAF/MEK inhibition to ICIs as a consequence of a limited intracranial response to BRAF/MEK inhibition. In patients receiving ICIs, the number of treatment changes after follow-up MRI scans decreased in time on treatment, probably due to the fact that immunotherapy treatment failure occurs in the first 6 months of ICI administration and the clinical condition remains stable beyond 6 months of therapy in the majority of patients [169]. Nonetheless, although MRI scans proved to aid in the early detection of melanoma brain metastases and support adaptative treatment strategy changes, more research is warranted to determine the impact and cost effectiveness of regular brain imaging and subsequent treatment changes in survival and to determine the optimal interval of screening and follow-up MRI scans.

At the moment, there are at least three MRI variants available for use in clinical settings, these include: (1) T2-weighted (T2w) imaging, which relies on the assessment of the differences on the T2 relaxation time of tissues; (2) diffusion-weighted imaging (DWI), which involves the calculation of an apparent diffusion coefficient (ADC) from DWI; and (3) dynamic contrast-enhanced (DCE) imaging, which measures T1 changes in tissues over time after the administration of a contrast agent [172]. The applications of those MRI techniques in dermato-oncology include the assessment of therapeutic responses in CM and SCC [173,174], differentiation of pseudoprogression from progressive disease [175], and in vivo assessment of tumor angiogenesis for tumor characterization and treatment planning [176]. Regarding the evaluation of therapeutic responses in skin cancer patients, the ADC calculated from DWI proved to be a reliable biomarker in this sense [177]. Measuring the ADC coefficient of water can reveal the status of the tumor tissue in that healthy tissue restricts water mobility due to the intact cell membrane, while the necrotic tissue exhibits higher membrane permeability, which results in an increased diffusion coefficient [178]. In parallel, DCE-MRI showed the most promising results when evaluating the pseudoprogression in melanoma patients with brain metastases who underwent immunotherapy [175]. DCE-MRI derived capillary permeability (Ktrans) and plasma volume measurements were inferior in CM patients experiencing pseudoprogression, containing the potential to serve as valuable biomarkers for disease monitoring in CM patients [175]. Nonetheless, the most appropriate technique for the in vivo assessment of tumor angiogenesis remains MR angiography. When coupled with specific contrast agents, MR angiography enables the evaluation of the morphologic structure of tumor vessels in relation to tumor vessel permeability, which is of great importance for patient surveillance and treatment guidance in clinical settings [179].

### 2.7. Near-Infrared (NIR) Bioimaging

In vivo near-infrared (NIR, 0.7–1.7 μm) bioimaging is rapidly expanding in the dermato-oncology field as it is a promising technique for the visualization, resection, and treatment of cancerous tissue. The NIR spectrum region has been divided into three channels, which include the NIR-I channel (0.7–0.9 μm), the NIR-IIa (1.3–1.4 μm) channel, and the NIR-IIb (1.5–1.7 μm) channel [180]. Fluorescence imaging in the NIR wavelengths between 700 and 900 nm emerged as a feasible optical approach that can provide larger imaging depth and better signal-to-noise ratio than traditional fluorescence bioimaging in the visible spectral region due to the low absorption of biological molecules in the NIR spectral region. Besides circumventing critical drawbacks such as photon scattering, photon absorption, and tissue autofluorescence, NIR fluorescence proved to be less hazardous to biological samples than the short-wavelength excitation light in visible fluorescence bioimaging, as it employs lower photon energy [181]. Yet, little attention has been paid to fluorescence imaging in the 900–1000 nm NIR-Ib window because water, the principal constituent of the biological samples, has an absorption peak centered at 970 nm. Thus, fluorophores with emission peaks in the NIR-Ib window are almost absent and most photodetectors of NIR light have a sensitivity restricted up to 900 nm [182]. Nonetheless, current studies with single-walled carbon nanotubes, quantum dots, and small molecules that fluoresce in the 1000–1700 nm NIR-II window highlighted superior spatial resolution, signal-to-background ratio, and penetration depth when imaging at longer wavelengths. Subsequently, these discoveries have led to a growth of interest in the application of NIR-II fluorescence imaging and the development of NIR-II fluorophores [182]. Currently, the only Food and Drug Administration (FDA)-approved NIR fluorophores for clinical use are the NIR-I dyes indocyanine green (ICG, emission at ∼800 nm) and methylene blue (MB, emission at ∼700 nm). The two dyes are primarily used for NIR fluorescence-based intraoperative imaging for structural visualization of anatomical features, such as the gastrointestinal tract, bile duct, and ureters [180]. Many other NIR fluorophores are currently under development and optimization for more specific anatomical and molecular imaging. For a long time, organic dyes such as squaraines, BODIPY, porphyrins, phthalocyanines, or cyanines seem to have been the first option as NIR contrast agents for biomedical imaging [183]. Yet, within recent years, many other types of NIR-II channel fluorophores, including lanthanide-based nanocrystals, fluorescent proteins, quantum dots, and carbon nanotubes with improved quantum yield, photostability, and biocompatibility have been developed, and tested in vivo for their utility in various clinical conditions, including those in dermato-oncology [180]. In particular, for skin cancers, NIR fluorescence imaging proved helpful for the identification of sentinel lymph nodes (SLNs) in CM [184] and for the visualization of fluorescence-labeled immune cells, especially to track and monitor them in living organisms during immunotherapy [185]. Despite its potential for clinical translation, NIR bioimaging still faces challenges and one of the most important is the lack of consensus in performing quality control and standardization of procedures and systems [186].

Standard management for CM patients with clinically node-negative tumors relies on the assessment of the regional lymph nodes by means of an SLN biopsy, which can provide significant information on melanoma staging and the most appropriate treatment. Lymphatic mapping is usually performed using a blue dye (isosulfan blue or methylene blue) and a radioactive tracer (technetium [Tc 99m]) to identify the first lymph node that receives direct lymphatic drainage from a primary tumor site [187]. Despite considerable advances in surgical techniques and medical technologies, false-negative SLN biopsy is still a challenge, being reported in approximately 17% of melanoma cases [188]. This is mainly due to the fact that both SLN mapping agents, Tc 99m and blue dyes, lack a strong optical signal-to-guide real-time intraoperative identification and hence present a limited target specificity. Lymphoscintigraphy with technetium Tc 99m may also present drawbacks such as poor spatial resolution (approximately 10 mm), elevated costs, logistical challenges, anaphylactic reactions, and potential skin necrosis among others [187]. Consequently, scientific efforts are now devoted to designing new optical imaging probes that can generate high-contrast and reliably localized SLNs and nodal micrometastases, enabling the real-time tailoring of surgical approaches while reducing the duration of anesthesia time and risk of nerve injury. A study led by Ferri et al. has recently highlighted that NIR fluorescence imaging using ICG may be a more effective, safe, less costly, and more convenient alternative for the identification of SNL in melanoma [189]. The authors presented the case of a 51-year-old patient with melanoma in his upper back, where NIR fluorescence was successfully employed to detect two SLNs in the left axilla. They also reported that ICG, a water-soluble tricarbocyanine dye, offers potential advantages for radiotracer-based lymphoscintigraphy including good tissue penetration, an excellent safety profile, and real-time intraoperative imaging capabilities. However, at the moment, NIR fluorescence using ICG is recommended for clinical use only in combination with lymphoscintigraphy for primary CM [189,190]. Nonetheless, other research groups have focused on examining the potential of nanoparticles in NIR imaging for the intraoperative detection and visualization of SLN. One such study was conducted by Zanoni et al. between February 2015 and March 2018 on 67 patients enrolled in a Phase II nonrandomized clinical trial at the Memorial Sloan Kettering Cancer Center (NCT02106598) [187]. The patients underwent preoperative localization of SLNs with technetium Tc 99m sulfur colloid, followed by a microdose administration of integrin-targeting, dye-encapsulated NPs, and surface modified with PEG chains and cyclic arginine-glycine–aspartic acid–tyrosine peptides (cRGDY-PEG-Cy5.5-nanoparticles) intradermally. No adverse events were observed following molecularly targeted core–shell silica NP administration. The concordance rate between preoperative lymphoscintigraphy with technetium Tc 99m and particle-based fluorescence-guided biopsy in detecting SLNs was 90% (95% CI, 74–98%) and five of the SLNs were metastatic. Moreover, the authors reported that NP-based fluorescence-guided SLN biopsy is well-tolerated and its increased sensitivity even in difficult-to-access anatomic sites may facilitate intraoperative SLN identification without the extensive dissection of adjacent normal tissue or nerves in patients with melanoma [187]. Nonetheless, although the use of cRGDY-PEG-Cy5.5-nanoparticles, in conjunction with real-time NIR intraoperative optical imaging guidance, proved to be a safe, reliable, and clinically feasible procedure for the accurate and sensitive detection of SLNs in patients with head and neck melanoma, further studies are required to standardize the technique and test the reproducibility of results in larger clinical cohorts.

It is also worth mentioning that NIR fluorescence may be harnessed to understand the function and underlying mechanisms of immune responses in CM patients subjected to immunotherapy, enabling the real-time in vivo monitoring and tracking of fluorescence-labeled immune cells of interest. The discovery of immune checkpoints and the development of ICI was a major breakthrough in cancer research that was undoubtedly fostered by a comprehensive understanding of the tumor microenvironment (TME) and its immunophenotype. ICI-based immunotherapy consists of neutralizing antibodies against negative regulators of immune function, such as cytotoxic T-lymphocyte-associated protein 4 (CTLA-4), programmed cell death protein 1 (PD-1), and PD-1 ligand 1 (PD-L1), thereby restricting the ability of residual tumor cells to escape the cytotoxic immune program [147]. The different stromal cells in the TME have important roles in shaping tumor evolution and therapeutic responses. The cells within the stromal compartment include (1) immune cells (macrophages, B lymphocytes, and T cells [e.g., CD4+ and CD8+ T cells], neutrophils, natural killer (NK), and dendritic cells (DCs)), (2) mesenchymal cells (fibroblasts, myofibroblasts, cancer stem cells (CSCs), mesenchymal stem cells (MSCs), adipocytes, and endothelial cells), and (3) myeloid derived suppressor cells (MDSCs) [147]. It is well known that an optimal antitumor immune response depends on a complex interplay between CD8 cytotoxic T cells, CD4 helper T cells, DCs, and NK cells [191]. Yet, in melanoma’s TME, TAMs are abundant and, due to their pro-tumoral M2 phenotype, these “tumor-hijacked” cells sustain therapy resistance [147]. In addition, through their secretome, regulatory T cells (also called Tregs), MDSCs, TAMs and tumor-associated stromal cells inhibit the anti-tumor T-cell response and maintain an immune tolerant tumor that attenuates the effectiveness of immune checkpoint inhibition [192]. Therefore, assessing the dynamics and activity of immune cells following immunotherapy through NIR imaging may be not only suggestive of therapeutic responses in CM patients, but also would support research into improving the clinical outcomes and gain mechanistic insights concerning immunotherapy resistance mechanisms in this hard-to-treat disease.

Although NIR fluorescence imaging seems to be the ideal tool for the observation of immune components in a real-time setting, the selection of the right fluorophores is a key step in this process, as the physicochemical properties of labeling agents (e.g., molecular weight (MW), absorption/emission wavelengths, surface charges, hydrodynamic diameter (HD), pKa, photostability, hydrophobicity, and plasma protein binding) may greatly impact their optical performance. The ideal imaging probe for labeling the immune components of interest should have the following essential properties: high imaging affinity and specificity for the desired immune components, acceptable safety profiles, and minimal immunogenic toxicity [193]. The labeling tools that are currently available for marking immune cells in the NIR window include small-molecule fluorophores (cyanines, phthalocyanines, porphyrin derivatives, squaraine derivatives, and BODIPY analogs), nanoparticles (nanocrystals, QDs, and metal nanoshells), and targeted (antibodies, peptides, and protein complex probes) and activatable probes [185]. Activatable probes encompass in essence the same compounds as targeted probes, but their fluorescence can be quantitated in response to internal microenvironment stimuli such as pH, enzymes, and oxidative stress [193]. NIR probes may be applied both in vivo and ex vivo for monitoring the behavior of immune components. Notably, ex vivo labeling is not restricted to effector immune cells and may be applicable to any type of immunological agents, including vaccines [194]. This method is frequently used for T-cell tracking in the context of cancer immunotherapy or Chimeric antigen receptor (CAR) T-cell therapy in preclinical models [193,195]. There are, however, significant limitations to the ex vivo method as the isolation and labeling procedures may affect the viability of lymphocytes. Therefore, a new methodology is highly desired to indirectly label sensitive immune cells such as T and B cells in vivo with exogenously administered fluorophores [185]. Currently, the scientific efforts for in vivo NIR imaging are focused on using specific fluorophores that can target particular surface receptors or membrane transporters of cells and on employing activatable fluorophores for monitoring and assessing the complexity of the immune responses in living organisms [193]. Several authors exploited the potential of peptide substrate that are responsive to granzyme B (GranB) for the in vivo evaluation of immunotherapy. GranB, released mainly by cytotoxic T (CTLs or CD8+ T) cells and natural killer cells during the cellular immune response is a valuable biomarker for immune activation in tumors [196]. He and colleagues synthesized two NIR macromolecular reporters engrafted on two different peptide substrates: CyGbPF and CyGbPP, containing N-acetyl-Ile-Glu-Phe-Asp (IEFD) and N-acetyl-IleGlu-Pro-Asp (IEPD), respectively, which are responsive to GranB for the real-time in vivo evaluation of immunotherapy [197]. The reporters are initially non-fluorescent due to the reduced electron-donating ability of the oxygen atom in CyOH. However, GranB released by CTLs in the TME triggers the cleavage of the peptide-caged moiety of the fluorophores, triggering the conversion of CyGbPF or CyGbPP into CyOHP, which displays an enhanced NIR fluorescence signal. Therefore, the signal intensity is linked with GranB levels in the TME, allowing for in situ evaluation of immunotherapeutic response in living animals [197]. Other authors exploited a BODIPY fluorophore conjugated with a prodrug (e.g., doxorubicin and DOX) that exhibits a dose-dependent turn-on fluorescence response and cytotoxicity in lipopolysaccharide-induced proinflammatory M1 macrophages for the real-time monitoring of targeted therapies including prodrug activation and intracellular trafficking in different cancer models [198]. Finally, there were also several reports that presented the efficiency of superoxide anion (O_2_−)-activatable NIR chemiluminescent reporters (SPNRs) to detect O_2_− for real-time in vivo NIR imaging of drug-induced cancer immune activation [199,200]. This mechanism relies on the principle that cytotoxic T cells have higher endogenous O_2_− levels than cancer and normal cells and that SPNR may link the chemiluminescence signals with the O_2_− levels to depict the populations of activated cytotoxic T cells and helper cells during immunotherapy administration [193].

In addition, immune cell-targeted NIR imaging could help solve one of the most challenging aspects of cancer immunotherapy, the “pseudo-progression”. Pseudo-progression consists of an initial increase in the size of tumor lesions observed in MRI or CT scans, followed by a delayed therapeutic response, leading to premature cessation of efficacious treatment owing to the false judgment of progression. Pseudo-progression has been reported in about 10% of immunotherapy treatment cases and is attributable to the recruitment of various immune cells (e.g., T and B lymphocytes) in the tumor and not due to tumor cell proliferation [4]. Although several procedures have been proposed to diagnose pseudo-progression after immunotherapy, such as MRI [201], ultrasonography [202], PET/CT [203], alongside blood parameters (LDH and S100) [204], circulating tumor (ct)DNA [4], and serum interleukin (IL-)8 level detection [205], the diagnostic accuracy for each of them remains questionable. Nonetheless, coupling NIR imaging with immune cell targeting fluorophores might help depicting the infiltration and recruitment of various immune cells in the tumor bed, which may be a valuable way to assess the therapeutic responses and differentiate pseudo-progression from true progression in cancer patients treated with immunotherapy.

## 3. Molecular Imaging Techniques

The field of molecular imaging, defined by the Society of Nuclear Medicine (SNM) as “the visualization, characterization, and measurement of biological processes at the molecular and cellular levels in humans and other living systems” has emerged as a vibrant field of research within recent years [206]. It includes two- or three-dimension imaging and quantification over time [207]. Nonetheless, molecular imaging techniques rely on the use of tracers to mirror molecular targets and pathways occurring within a particular area of interest, in contrast to conventional imaging techniques that differentiate qualities, such as density (CT), water content (MRI), or reflectance (US) [207]. The most important molecular imaging techniques are the single-photon emission computed tomography (SPECT) and positron emission tomography (PET) imaging modalities, which have become essential in the clinical decision-making process in various pathologies, including skin cancer [206]. PET imaging relies on the unique properties of radioactive isotopes that decay via positron emission. In contrast, SPECT imaging detects gamma-ray photons emitted during their radioactive decay by rotating the camera around the subject to capture the gamma emissions in 3D [208]. The latest advances in technology and hardware detection systems, such as new crystals and digital detectors, have also fostered the rapid evolution and implementation of these two techniques in the clinical setting. Nonetheless, both SPECT and PET are nuclear methods of molecular imaging, which means that they rely on the use of radionuclides as contrast agents. Contrast agents are essential for nuclear imaging and the radionuclide is usually attached to a targeting vector. The radionuclides applied as contrast agents in SPECT imaging include ^123^I, ^99m^Tc, and ^111^In, whereas ^68^Ga, ^18^F, and ^64^Cu are the most suitable for PET imaging [208]. To obtain a molecular signature of the disease in a non-invasive fashion, the specific accumulation of PET/SPECT radiotracers at the disease sites is highly desirable. Targeting vectors include but are not limited to peptides, nucleotides, antibodies, and nanoparticles. These targeting vectors are designed to target specific cancer biomarkers or biological processes related to tumorigenesis (e.g., hypoxia, acidosis, and energy metabolism) and hence dictate the biodistribution of the radioactive molecule within the body. Promising targeting vectors display high binding affinity to the specific receptors expressed on target cells, rapid clearance from nontarget tissues, in vivo stability, and low immunogenicity or toxicity [209]. Nonetheless, within recent years, the search for more sensitive and specific targeted vectors that can meet the clinical needs of radionuclide imaging has become an active area of research.

Antibodies have long circulating half-lives and require between 4 and 7 days to obtain high tissue contrast and optimally image the targets of interest. As it may delay the administration of an alternative treatment, the relatively longer biological half-lives of radiolabeled antibodies pose important limitations to the clinical workflow [210]. The long-circulating half-lives also result in the need for long-lived radioisotopes and, hence, more prolonged radiation exposure to patients [21]. In addition, the size of antibodies may restrict their diffusion into solid tissues where cells expressing particular biomarkers might be found. The delivery of antibodies to tumor cells is a complex task with many factors involved, such as blood vessel density, interstitial fluid pressures, vascular permeability, and tumor growth kinetics, and these variations usually result in suboptimal PET scans [211]. Finally, antibodies contain Fc domains that trigger antibody-dependent cellular cytotoxicity (ADCC), and—if binding to PD-1 or CTLA-4 on T cells—they may orchestrate the intratumoral depletion of T cells mediating the antitumor response in human cancers [212].

To address the problems arising from the use of antibodies, scientists have developed novel probes with shorter biological half-lives and lower molecular weight. Specifically, small peptides and low-molecular-weight imaging agents present faster pharmacokinetics (from minutes to hours) than antibodies and produce superior image contrast and higher tumor-to-normal tissue ratio. Particularly for PET, the most routinely used clinical radiopharmaceuticals are either peptides or low-molecular-weight agents [210]. Peptides present many advantages, such as high affinities and specificities for their targets, fast clearance from blood and other non-target sites and good biocompatibility, resulting in excellent tumor-to-background ratios. Moreover, they show superior tissue/tumor penetration capabilities with lower immunogenic effects than antibodies and intermediate in vivo stability. Therefore, peptides have become one of the most suitable and practical platforms for radionuclide labeling in medicine. Finally, peptides present increased flexibility for chemical modifications, which are expected to enhance peptides’ stability and practical properties and expand their application in preclinical and clinical studies [213].

The development of SPECT and PET imaging, combined with specific targeted radiolabeled molecules, has opened a new era of clinical imaging. However, PET and SPECT lack spatial resolution and are often coupled with computed tomography (CT) or magnetic resonance imaging (MRI), which provide an exquisite spatial resolution of anatomical structures and help remove nuclear imaging data artifacts using attenuation correction. Therefore, dual-modality imaging systems combine the advantages of these two imaging techniques and provide a more comprehensive view of the pathobiology of tumors [208,214]. Here, we summarize the available scientific literature focused on the applications of PET/SPECT in skin cancer and provide an overview of the conducted research within the past decade, with an additional focus on the functional nuclear imaging agents and novel mechanistic targets in this disease.

### 3.1. Single Photon Emission Computed Tomography (SPECT)

SPECT is a technique that has roots in the pioneering experiments with emission tomography performed by Kuhl and Edwards in 1963. Although the rotating gamma camera systems have dominated the clinical setting for a long time, the latest technological innovations have enabled the development of organ-specific dedicated SPECT systems, which have become the preferred option for many clinicians [215]. Nonetheless, the most important application of SPECT/CT in clinical practice remains to be preoperative lymphatic mapping, especially in tumors such as melanoma, cervical cancer, breast cancer, and head and neck carcinoma [216,217]. SPECT/CT may help identify SNs in the case when they are located deep below the surface or near the injection site or when the lymphatic drainage is unclear. Moreover, SPECT/CT may discriminate precisely the activity arising from two or more lymph nodes placed in the vicinity of one another that are usually depicted as a single hotspot on planar images. As a consequence, accurate SN localization in a preoperative setting facilitates surgical access and improves overall detection and survival rates in the affected patients [218]. For these reasons, SPECT/CT is considered a valuable complementary approach to planar lymphoscintigraphy for SN identification. A combined interpretation of both modalities hold promise to improve the standard care for SN detection in melanoma, as it offers the possibility of detecting even the second lymph node level [219].

Van der Ploeg and his team were among the first research groups aiming to investigate the efficiency of SPECT/CT technique in the detection of SNs in melanoma patients with inconclusive conventional lymphoscintigrams. They observed that SPECT/CT has added value in 30 out of 85 patients (35%), by detecting additional SNs not previously visualized on planar images or by modifying the surgical incision planning [220]. In line with these observations, a multicenter study conducted on 1508 cancer patients (1182 breast cancer, 262 melanoma, and 64 pelvic cancer (prostate, cervix, penis, vulva) patients) reported that SPECT/CT improved the detection of SNs when compared with planar lymphoscintigraphy, in a considerable number of patients in all the malignancies studied [221]. The authors further recommended to perform SPECT/CT in all the patients with melanoma of the head and neck or trunk, all the patients with pelvic malignancies, and those with breast cancer and melanoma patients with unexpected drainage on planar lymphoscintigraphy. In addition, SPECT/CT data changed the drainage territory, leading to surgical adjustments in almost 37% of the melanoma patients [221]. In parallel, another clinical study on 18 metastatic melanoma patients described a visualization rate of 100% for SPECT/CT versus 89% for planar lymphoscintigraphy; notably, SPECT/CT provided additional anatomical information leading to an adjustment of the surgical approach in four patients [222]. Interestingly, one study showed that although the addition of SPECT/CT to the SNB in patients with CM results in a certain increase in the cost of the preoperative imaging procedure, it may decrease by 30% the expenditure associated with operative time, hospital stay duration, treatment of advanced-stage disease, and the use of general anaesthesia [223]. Finally, there are also several reports highlighting a significant therapeutic benefit in terms of improved disease-free and disease-specific survival in melanoma patients undergoing SPECT/CT imaging for SN staging [218]. However, the improved accuracy comes with an increased workload for pathology departments and hence an increased risk of cancelation of the SNB procedure on the day of surgery, which may negatively impact nodal relapse-free survival [224].

The introduction of SPECT/CT has genuinely revolutionized the preoperative lymphatic imaging scenario in cancers; moreover, it disseminated rapidly into the hospital environment due to the attainability of Tc 99m generators with reasonable cost-effectiveness, which allows for the in-house formulation of the desired radioactive products. [225]. In a similar fashion, by relying on the accessibility of fluorodeoxyglucose (^18^F-FDG), PET/CT rapidly entered into the clinical practice for various oncological applications [21]. Nonetheless, although PET/CT images are inherently superior to SPECT/CT images, this improvement involves a cost significantly higher. As a result of recent advances in detector technologies and informatic processing algorithms, the spatial resolution of SPECT images is expected to approach that of PET images, without experiencing any decrease in sensitivity. Yet, the answer to the question concerning which technology will dominate the nuclear imaging field remains elusive [226]. At a glance, when compared to SPECT/CT, PET technology provides improved resolution and less attenuation and scatter artifacts and, therefore, superior imaging capabilities. In order to achieve an increased sensitivity, PET generally operates with more versatile and reliable tracers, which come with exorbitant costs limiting the availability of PET imaging. In contrast, radiopharmaceuticals needed for SPECT are cheaper, have increased availability, and, due to the longer half-life of single photon emitters, display more effective targeting abilities of molecular targets of interest, allowing for an accurate imaging of the biological processes at equilibrium in vivo [226].

### 3.2. Positron Emission Tomography (PET)

PET is one of the most sophisticated molecular imaging techniques that may reliably help visualize malignant lesions in structures that macroscopically look normal. For a more exact diagnosis, PET scans necessitate injecting a small quantity of bioactive molecules such as glucose or oxygen, labeled with radionuclides such as ^11^C, ^18^F, ^13^N, or ^15^O. PET tracers will distribute in the body upon intravenous injection when they can bind to predetermined targets in patient tissues [227]. Imaging with PET can provide real-time information on different biological processes such as glucose metabolism [228], DNA replication, hypoxia and inflammation [229], or gene expression based on interaction events between the specific PET tracer and cellular enzymes [230] or can provide details about the distribution and density of tumor receptors or other binding targets [231]. In the clinical setting, biological information obtained from PET imaging is widely used in diagnosing disease, monitoring response to therapy and minimal residual disease (MRD), and aiding in personalized medicine [21]. Therefore, it provides mechanistic insights concerning tumor heterogeneity and prognosis and identifies instances of treatment failure in various tumors, including skin cancer. It is also worth mentioning that PET remains the most popular technique for advanced CM staging and therapy response evaluation [232]. Table 1 depicts the advantages and limitations of the PET technique compared with other imaging techniques available for skin cancer imaging.

The increased metabolism of glucose in viable cancer cells causes the glucose analog ^18^F-fluorodeoxyglucose (^18^F-FDG) to be the most commonly used PET radiotracer in clinical practice [243]. Similar to glucose, FDG is transported into cancer cells by glucose transport proteins such as GLUT1. Once transported into the cell, FDG is phosphorylated by the glycolytic enzyme hexokinase to become ^18^F-FDG-PO4, which is retained within the tumor cell because of the low membrane permeability of FDG-6-phosphate and used for generating tumor contrast. Given that the glucose metabolism is manifold increased in malignant cells than in normal tissues, ^18^F-FDG PET is the preferred technique for cancer imaging in oncology [244]. Nonetheless, ^18^F-FDG PET/CT plays a limited role in staging early melanoma (AJCC stages I and II); still, it proved more useful in the initial staging of advanced cutaneous melanoma (AJCC stages III and IV) and assessment of disease recurrence, demonstrating a high accuracy in the detection of lymph node, soft tissue, and visceral metastases [241]. It was found that among sonography, CT, PET, and PET/CT, US is the most efficient for node staging, whereas PET/CT is more appropriate for the detection of distant metastases (80% sensitivity and 87% specificity) [245]. The usefulness of ^18^F-FDG PET/CT in the detection of distant melanoma metastases was also confirmed by another study [246]. However, increasing evidence suggests that PET/CT may not be sensitive enough in assessing melanoma brain metastases, which are one of the most life-threatening complications of the disease. A suggestive example in this context is the study conducted by Aukema et al., which aimed to assess the diagnostic values of ^18^F-FDG PET/CT and MRI in melanoma patients with palpable lymph node metastases [247]. The authors found that ^18^F-FDG PET/CT changed the intended regional node dissection in 26 patients (37%), resulting in a superior diagnostic accuracy of 93%. Still, they missed five patients with brain metastases that were detected only by MRI [247]. Several reports also documented the lack of accuracy of ^18^F-FDG PET in detecting metastatic lesions of less than 1 cm in the brain, liver, or lung, suggesting that the sensitivity of PET methodology in detecting regional metastases may also be influenced by the tumor volume [248,249]. Crippa et al. showed that FDG-PET is able to detect 100% of metastases ≥10 mm, 83% of metastases 6–10 mm, and 23% of metastases ≤ 5 mm. Furthermore, the authors emphasized that FDG-PET presented high sensitivity (greater or equal to 93%) when operating with metastases with more than 50% lymph node involvement or with capsular infiltration but displayed a limited sensitivity in detecting subclinical microscopic disease [250]. These findings substantially align with the observations of Schröer-Günther and colleagues, who conducted a meta-analysis regarding the usefulness of ^18^F-FDG PET/CT for CM staging on 17 clinical studies encompassing patients at the I–IV AJCC stages [251]. Interestingly, the authors found that PET/CT’s sensitivity and specificity was significantly higher (sensitivity 68–87%, specificity 92–98%, respectively) in CM patients at stage III and IV with respect to CM at an earlier stage. In parallel, ^18^F-FDG PET/CT was reported to be less sensitive with respect to lymphoscintigraphy for the detection of nodal metastases in patients with AJCC stage I–II, therefore PET/CT should not replace standard procedures such as lymphoscintigraphy and SNB for the initial staging and stratification of patients with early stage disease [251]. In light of the aforementioned results, the use of a conventional PET/CT approach might be considered adequate for advanced stage disease or in case of doubtful findings at conventional imaging.

PET technology allows not only for qualitative image interpretation but also for the calculation of several quantitative parameters related to metabolic activity, such as maximum and mean standardized uptake value (SUVmax and SUVmean) and metabolic volume, such as metabolic tumor volume (MTV) and total lesion glycolysis (TLG). TLG is the product of MTV and SUVmean [252]. Of note, MTV and TLG were found to be suggestive of tumor aggressiveness and poor clinical outcomes in CM patients. For instance, Son et al. observed that MTV and TLG may be valuable prognosticators of clinical outcomes among patients with primary cutaneous melanoma [253]. In a similar fashion, Reinert et al. noticed that MTV and TLG of the primary tumor are independent prognostic biomarkers for unfavorable disease evolution in a clinical cohort of 107 advanced CM patients undergoing ^18^F-FDG PET/CT for planning metastasectomy. Notably, the authors found strong correlations between volumetric PET parameters and the established serologic tumor biomarkers such as lactate dehydrogenase (LDH), S-100 protein, and inflammatory markers alkaline phosphatase (AP) and C-reactive protein (CRP), highlighting that tumor volumetric parameters may have important prognostic applications in clinical management of CM patients [254].

Besides its usefulness in CM staging and prognosis, FDG PET has largely been used for evaluating the therapeutic responses in patients receiving systemic therapy, especially ICIs [255,256]. One study conducted by Ito et al. reported that whole-body MTV obtained from baseline PET/CT scans might be a valuable prognostic biomarker among other clinical prognostic factors in melanoma patients treated with ipilimumab [257]. Since immunotherapies work by reactivating the immune system, they may often trigger unconventional response patterns and immune-related adverse effects in CM patients. Because of the inflammatory response induced by ICI, neoplastic lesions may appear stable or even more prominent in size and metabolic activity before effective shrinkage, causing it to be challenging to discriminate between progression and pseudo-progression via imaging [258]. The Response Evaluation Criteria in Solid Tumors (RECIST) V.1.1 guidelines are the mainstay for assessing therapeutic responses in clinical trials involving patients with solid tumors receiving anti-cancer therapy; however, RECIST 1.1 does not capture atypical response patterns, such as pseudo-progression, that are observed in a considerable percentage of individuals who receive ICI treatment. Hence, RECIST 1.1 considers pseudo-progression to be progressive disease, which often results in the discontinuation of an effective treatment [259]. Numerous response criteria have been proposed to overcome these limitations, including immune-related response criteria (irRC) and immune-modified (im)RECIST, in which the appearance of new lesions is not always synonymous with the progression of the disease, requiring confirmation at least after 4–8 weeks. In a similar fashion, metabolic criteria based on ^18^F-FDG PET/CT have been modified to improve diagnostic accuracy during immunotherapy [260]. Therefore, as proposed by PET Response Evaluation Criteria for Immunotherapy (PERCIMT) and Immunotherapy-modified PET Response Criteria in Solid Tumors (imPERCIST) criteria, respectively, according to their number and size, new lesions may be considered a sign of progression if the metabolic activity is greater than a predetermined cut-off [260]. A relatively recent study by Rivas et al. showed that the evaluation of ICI responses using metabolic imaging with imPERCIST5 and PERCIST5 (criteria used for conventional chemotherapy) was significantly associated with overall survival in 29 patients with advanced or metastatic cutaneous melanoma [261]. Among the many variants of SUV (e.g., mean SUV, maximum SUV), SUV corrected for lean body mass (SUL) was chosen for use with PERCIST because it is less susceptible to variations in patient body weight than the other SUV metrics [262]. PERCIST specifies that the SUL peak should be obtained on the single most active lesion on each scan. The SUL peak is the average of the activity within a spherical region of interest, measuring 1.2 cm in diameter (for a volume of 1 cm^3^), centered on the most active part of the tumor [263]. Using a concept similar to RECIST, PERCIST recommends the sum of the activity of up to five target lesions (no more than two per organ) to be assessed as a secondary determinant of response. Nonetheless, in PERCIST, new lesions define progressive metabolic disease only if the SUL peak is greater than or equal to 30%, with at least a 0.8-unit visible increase in the extent of FDG uptake [263].

Although FDG-PET may offer valuable information on primary staging, response assessment post-therapy, staging of suspected recurrence, and prognosis, the results of PET examinations should be interpreted with care, as there are many aspects that may impact their accuracy.

Imaging pitfalls are considerably affected by several factors, including patient preparation, lesion characteristics as well as the preferential bio-distribution of ^18^F-FDG in certain tissues or organs. Pitfalls dramatically impact scan sensitivity and specificity, as they may result in false positives or negatives [241]. False-positive findings may result in unnecessary and invasive procedures; yet, false-negative scans may be associated with delayed diagnosis and treatment, thereby dramatically affecting the patient and clinician. False-positive results may occur due to the increased FDG uptake in some normal body areas, such as urinary tract structures, lymphoid tissue, and brown adipose tissue [242]. False positives may also be determined by the increased accumulation of FDG in some benign processes such as inflammation or infections. Post-inflammatory changes around the site of a surgical resection, benign conditions, bone fractures and degenerative bone disorders, thyroiditis, esophagitis, and pancreatitis display increased ^18^F-FDG uptake and can be easily confounded with metastases [242]. Moreover, FDG PET seems inappropriate for assessing brain melanoma metastases due to high physiologic ^18^F-FDG metabolism in those tissues [241]; yet, ^18^F-FDG PET/CT displays a reduced sensitivity for detecting small metastatic lung lesions due to partial volume effect, leading to underestimation of the tracer uptake [264]. Given these limitations, brain MRI and lung CT imaging should remain the primary detection methods for melanomas spreading to those secondary sites. Further complicating this scenario, metastases from different types of melanomas (e.g., cutaneous and uveal) demonstrate different degrees of ^18^F-FDG avidity, creating challenges for the appropriate management of melanoma patients. For instance, liver metastases from cutaneous melanoma are reliably ^18^F-FDG-avid, whereas those from uveal melanoma show low ^18^F-FDG avidity, resulting in false negative assessments [265]. Finally, PET-CT imaging involves the administration of radiotracers for image acquisition and is inherently linked to exposure to a low dose of ionizing radiation, presenting a small but not negligible risk to the patient. Compared to other cancers, melanoma affects younger patients; therefore, the cumulative risk of radiation exposure on lifelong surveillance over treatment is high and its associated effects should be strongly considered when choosing this imaging modality for patient monitoring [21]. Because of these limitations, there is an ongoing need to develop novel imaging modalities that are highly sensitive and specific in detecting melanomas in the affected patients.

### 3.3. Biological Targets for Cutaneous Melanoma Molecular Imaging

Most of the molecular imaging studies in skin cancer focus on melanoma since it is the most metastatic and deadliest disease among all the other cutaneous subtypes. The biochemical processes and molecular targets that are currently under investigation for melanoma imaging are associated with the immune response (e.g., PD-1/PD-L1 [266], indoleamine 2,3-dioxygenase (IDO) [267]), TME dynamics (e.g., fibroblast activation protein (FAP) [268]), melanomagenesis (e.g., melanin [269]), invasion (e.g., melanocortin receptor subtype 1 (MC1R) [270]), or angiogenesis (e.g., integrins [271]) and hence many other novel antibodies and angiogenesis-related molecules continue to reside in the arena of research and development. Here, we will focus on the most well characterized targets that showed promise for use in SPECT/PET imaging.

#### 3.3.1. Imaging the Immune Environment

As newer immunotherapies were introduced into the clinical practice, the necessity to non-invasively assess the changes in the immune system emerged as a critical necessity in the field. Immuno-PET, which combines the superior targeting specificity of monoclonal antibodies (mAb) with the outstanding sensitivity and resolution of PET, represents a new addition to the molecular imaging portfolio that seems appropriate for investigating the immune function in living organisms [272]. The cellular composition, dynamics, and spectrum of bio-active constituents within the TME can all play a vital role in the efficacy of immunotherapy and provide additional targets for pharmacological interventions. Therefore, it is becoming highly attractive to image these aspects of the TME to identify the mechanisms of immune evasion operative in each patient and guide the development of targeted therapeutic approaches that will increase the chances of therapeutic success in the affected patients [273]. To achieve this goal, several radiotracers have been developed to image interleukins [274] and specific immune cells, including B cells [275], NK cells [274], macrophages [276], myeloid cells [277], and T cells [278]. We are summarizing herein most recent evidence of immunoPET strategies in delineating T cells using immune checkpoint molecules, such as indoleamine 2,3-dioxygenase (IDO) and the PD-1/PD-L1 axis.

##### Indoleamine 2,3-Dioxygenase (IDO)

PET imaging may be used to evaluate the immunosuppressive signals within the tumor microenvironment. IDO1 (indoleamine 2,3-dioxygenase 1) is an enzyme involved in the degradation of tryptophan to kynurenine, which acts as a major inhibitor of the immune response. Decreased extracellular tryptophan levels and increased kynurenines levels inhibit T and NK cell proliferation and activation, mediating an immunosuppressive tumor environment whereby tumors can evade the immune system. Therefore, imaging the IDO-mediated kynurenine pathway of tryptophan metabolism with positron emission tomography (PET) could provide valuable information for the noninvasive assessment of the cancer immunotherapy response [279]. The increased expression of IDO is also associated with poor clinical outcomes and several IDO inhibitors are now being studied in clinical trials as adjuvants in the clinical management of a variety of cancers [267]. There are two IDO1 inhibitors in late-stage clinical trials and many others in earlier development. Epacadostat (Incyte) in Phase II clinical studies had a 58% overall response rate (ORR) in combination with Pembrolizumab for melanoma [280] and Indoximod (NewLink Genetics) in Phase II had a 45% ORR in combination with Gemcitabine and Abraxane for chemonaive pancreatic cancer (NCT02077881) [281]. Finally, the effectiveness of epacadostat (IDO inhibitor) and pembrolizumab was recently investigated in a phase III randomized study on 706 patients with unresectable melanoma (ECHO-301/KEYNOTE-252) [282]. Notably, no significant differences in terms of progression-free survival (PFS) or overall survival (OS) were reported between the patients treated with epacadostat plus pembrolizumab versus those receiving placebo plus pembrolizumab, suggesting that the usefulness of IDO1 inhibition as a strategy to enhance anti-PD-1 therapy activity in cancer remains highly controversial [282].

Within the last decade, several tracers for assessing expression levels of IDO1 in cancers have been developed. All these ligands are ^18^F or ^11^C-labeled derivatives of L or D isomers of tryptophan, except for an ^18^F-labeled epacadostat analog developed by Huang and his team in 2017 [267]. Among all those IDO1 imaging agents, α-[^11^C]methyl-L-tryptophan ([^11^C]AMT) remains the most important and appropriate for use in clinical settings. In patients with brain tumors, [^11^C]AMT-PET demonstrated increased tracer uptake in all grade II–IV gliomas and glioneuronal tumors compared to normal cortex [283]. In addition, several other [^11^C]AMT-PET studies demonstrated prolonged tracer retention and high uptake in xenograft mouse models of glioblastoma and metastatic brain tumors (from lung and breast cancer), highlighting the utility of this PET agent in the detection of these types of tumors in humans [284]. Further clinical studies with [^11^C]AMT-PET demonstrated the usefulness of this technique for visualizing and quantifying intratumoral tryptophan uptake in glioblastoma patients treated with an IDO1 pathway inhibitor, which may help delineating between true progression and pseudoprogression in patients receiving such therapy [285]. Finally, there are also underway several phase II clinical studies investigating ^11^C-AMT as a predictive imaging biomarker of response to pembrolizumab in patients with PD1 inhibitor-naïve metastatic melanomas (NCT03089606) [286]. For instance, Oldan et al. have showed that baseline ^11^C-AMT PET imaging using simple radiomics measurements (highest metabolic activity, SUVmax, tumor heterogeneity, and skewness) may better predict the clinical benefit from pembrolizumab in metastatic melanomas than FDG-PET [287]. Nonetheless, variability in ^11^C-AMT’s SUV_max_ cannot be solely explained by FDG-PET’s SUV_max_, suggesting that these two imaging modalities should complement each other to provide valuable information about intratumoral metabolic dysregulation that may relate with pembrolizumab response in melanomas [287].

##### PD-1/PD-L1

In many cancers, including skin cancer, PD-L1 expression has been positively associated with response to immunotherapy; therefore, treatment strategies are often guided by immunohistochemistry (IHC)-based diagnostic tests assessing the expression of PD-L1 [266]. Nonetheless, IHC requires an invasive tissue biopsy and hence is prone to a number of preanalytical (e.g., fresh vs. archival tissue samples) and analytical (e.g., different staining patterns between IHC assays) bias that may affect the PD-L1 assessment. Imaging can overcome these limitations, enabling a real-time assessment of the TME, particularly after treatment and when the location of the tumor is inaccessible [266]. Radiolabeled large (e.g., antibodies or their fragments, minibodies, and affibodies) and small (e.g., peptides and non-peptides) molecules have recently emerged as promising radiotracers for the imaging PD-1 or PD-L1 levels in cancer patients because they may guide therapy decisions and help in monitoring patients under immunotherapies [288]. Several clinical trials evaluating the potential roles of PD-(L)1 PET tracers in assessing PD-(L)1 expression in human cancers are recruiting or active [289,290]. Although trials were initially undertaken in NSCLC, the extensive use of ICI in other tumor types enabled the dissemination of PD-(L)1 PET tracer studies into SCC of the head and neck [291], renal cell carcinoma [292], Hodgkin lymphoma [293], breast cancer [294], CM [295], and other cancers. In CM, several studies have shown that certain immunoPET tracers such as ^64^Cu-DOTA-anti-PD-1-mAb or ^64^Cu-NOTA-PD-1-mAb may be successfully used for the non-invasive imaging of PD-1/PD-L1 expression and for examining the extent of tumor-infiltration of lymphocytes in tumor tissues [296,297]. Hence, Natarajan et al. reported that the greatest uptake of ^64^Cu-DOTA-anti-PD-1-mAb was achieved at 48 h post-injection, with a tumor-to-muscle uptake ratio of 11 in mice bearing B16-F10 melanoma tumors [297].

Many antibodies and derived minibody-, affibody-, and nanobody-based radiotracers have been developed for PD-1/PD-L1 imaging, all exhibiting acceptable accumulation in tissues [267,288]. Still, they possess increased immunogenicity and may potentially trigger adverse immunological effects such as cytokine storms in living organisms [288]. As a consequence, scientists reconsidered their use and are currently focusing on employing peptides and non-peptide small molecules in imaging studies, as they can ensure a higher tissue penetration and more rapid targeting of inaccessible tumor sites compared with larger molecules [298]. Moreover, these small molecules possess advantageous pharmacokinetic properties, a fast tissue uptake, quick clearance times, and a high signal-to-noise ratio that allows for imaging immediately after the injection, resulting in a reduced radiation exposure for the patient [288]. The biological half-lives of small peptides are similar to those of short-lived PET nuclides (^15^O, ^13^N, ^11^C, ^124^I, and ^18^F), which, especially in the case of ^18^F, enhance the resolution of the PET scans [288]. Feng and his team have recently filed a patent for the development of several radio-iodinated small molecule PD-L1 inhibitors that may be exploited for the imaging and treatment of PD-L1-positive tumors [288].

Besides the targets mentioned above, many other immune checkpoints are also investigated for cancer imaging. Imaging CTLA-4 and its ligands CD80 and CD86 emerged as an active area of research due to the FDA approval of the monoclonal antibody ipilimumab [267]. Currently, a first clinical CTLA-4 imaging study is ongoing, where tumor lesion uptake and the biodistribution of 89Zr-labeled ipilimumab will be assessed at the start of ipilimumab therapy and after the second injection three weeks later. In addition, this study aims to determine a putative correlation between tumor targeting of ipilimumab, response to treatment and the uptake in normal tissues, and elucidate the relationship between organ targeting and toxicity (NCT03313323) [299]. LAG-3 and TIM-3 are other important immune checkpoints that play significant roles in T-cell activation, proliferation, and homeostasis and may be harnessed for molecular imaging in tumors [273]. Multiple clinical trials are currently ongoing to evaluate the safety and efficacy of pharmaceuticals targeting LAG-3 in clinical cohorts (NCT05346276, NCT04566978) [300,301]. There is also an ongoing single-center, open-label clinical trial aiming to assess the safety and pharmacokinetics of the PET tracer 89Zr-DFO-REGN3767 directed against LAG-3 in 38 patients with advanced solid cancers before and during treatment (NCT04706715) [302].

As novel immune pathways and targets are identified for cancer therapy, additional radiotracers will likely be evaluated to image them. Reinforced by the recent advancement in radiomics and AI and coupled with the exquisite spatial resolution provided by CT, immuno-PET is expected to soon become a powerful tool that may offer outstanding insights to cancer biology [303].

#### 3.3.2. Imaging Integrins

The noninvasive determination of integrin expression is another exciting approach to melanoma detection. Integrins are heterodimeric glycoproteins consisting of α- and β-subunits, which regulate cell–cell and cell–extracellular matrix (ECM) interactions. There are 24 distinct combinations of the eight β-units and the eighteen α-units known [304]. The integrins expressed on endothelial cells are involved in the regulation of cell migration and survival during angiogenesis. In contrast, integrins expressed on carcinoma cells augment metastatic programs by facilitating invasion and movement across blood vessels [17]. The α_v_β_3_ integrin, which binds to Arginine-Glycine-Aspartic acid (RGD)-containing components of the interstitial matrix (e.g., fibronectin, thrombospondin, and vitronectin), emerged as one of the most important target structures in cancer imaging [271,305]. By activating and controlling multiple oncogenic pathways, integrin α_v_β_3_ expression enables melanoma cells to switch from a sessile, stationary state to a migratory and invasive phenotype [306].

Nonetheless, the scientific advances in the field culminated in the development of several radiolabeled RGD peptides targeting integrin α_v_β_3_ among specific clinical trials. Starting from the iodinated derivatives that have dominated the last decades, a variety of compounds labeled with almost all the available isotopes in nuclear medicine have been introduced [304]. The first and best characterized one is [^18^F]Galacto-RGD, which showed specific tumor uptake with fast clearance, mainly by renal excretion, resulting in good tumor-to-background ratios and low radiation exposure for the patient. Nonetheless, the radiosynthesis of this compound is complex, laborious, and extremely expensive [304]. Consequently, enormous efforts have been carried out to improve the radiolabeling strategy and develop even more effective RGD peptides for PET/SPECT imaging. One approach was focused on alternative ^18^F-labeling strategies, such as click chemistry, isotopic exchange labeling, and the introduction of aluminum fluoride species, among others. Another approach to developing new PET tracers focused on substituting ^18^F with ^68^Ga. Except for the isotopic exchange labeling strategy, all the other labeling strategies generated promising RGD peptides that entered into clinical studies and were associated with shorter production times as described for [^18^F]Galacto-RGD, with the most significant reductions found for the ^68^Ga-labeling procedure [304]. In addition, it was reported that the cyclization of RGD can enhance biological activity and significantly improve its selectivity and binding ability to receptors [307]. For instance, the cyclic Arg–Gly–Asp–D–Phe–Lys (cRGDfK) sequence presents a higher binding affinity to integrin α_v_β_3_ than the linear RGD peptide due to its ability to interfere with the adhesion of fibronectin to cells. Hence, the cyclic conformational structure may also prevent proteolysis [308].

The first approaches to introduce ^68^Ga-labeled RGD peptides were focused on the use of DOTA-conjugated RGD peptides. Decristofor et al. have compared the diagnostic efficacy of ^18^F-Galacto-RGD with that of ^68^Ga-DOTA-RGD and ^111^In-DOTA-RGD and found that tumor uptake ratios were comparable for all the three agents in melanoma mice models [309]. Nonetheless, the authors reported that due to its lower blood pool activity, [^18^F]Galacto-RGD should remain the main option for imaging α_v_β_3_ expression in cancer [309]. Although ^68^Ga-DOTA-RGD showed promising results for integrin imaging, it is known that the cyclododecane ring of DOTA does not have the optimal size for complexing gallium. Therefore, a more favorable chelating system seems to be the NOTA system, which contains a nine-membered ring more suitable for binding ^68^Ga [304]. This system was then used in conjunction with NOTA-RGD [310] and NODAGA-RGD [311]. As displayed by [^18^F]Galacto-RGD, the latter peptide presented significantly reduced binding to plasma proteins compared to [^68^Ga]DOTA-RGD, resulting in similar imaging properties in a murine tumor model [304]. In recent years, alternative chelating systems have been introduced for ^68^Ga-labeling of RGD peptides. This includes RGD peptides conjugated to H_2_dedpa derivatives and TRAP(RGD)_3_ [304]. Notni et al. designed ^68^Ga-avebetrin (also known as ^68^Ga-TRAP(RGD)_3_) and compared the PET data of ^68^Ga-TRAP(RGD)_3_ with that of ^68^Ga-NODAGA-c(RGDyK) and ^18^F-Galacto-RGD in terms of tracer biodistribution. In contrast with ^68^Ga-NODAGA-RGD and ^18^F-Galacto-RGD, ^68^Ga-TRAP(RGD)_3_ displayed rapid blood clearance and renal excretion while maintaining an optimal concentration in tumor tissues, which indicates its potential as a promising α_v_β_3_ imaging agent in cancers [312]. A recent addition to the panel of ^68^Ga-labeled RGD peptides is [^68^Ga]NOPO-RGD. The NOPO chelator belonging to the “TRAP family” showed excellent ^68^Ga labeling properties even in the presence of high concentrations of competing metal cations, serving as a foundation for the synthesis of novel radiotracers with improved imaging capabilities [304,313]. Indeed, this discovery facilitated the design of the ^68^Ga-NOPO–c(RGDfK) probe, which exhibited a higher degree of hydrophilicity than similar conjugates with other chelators, resulting in the rapid and specific uptake in M21 tumor xenografts, very rapid pharmacokinetics, and renal clearance [313].

Among the ^18^F labeling strategies of peptides, the Al^18^F labeling method has recently gained considerable interest as it can be used with many targeting molecules (e.g., small molecules, peptides, proteins) and hence retain high binding affinities when complexed with the NOTA ligand [314]. The ^18^F-labeled dimeric RGD-peptide [^18^F]AlF-NOTA-PRGD2 ([^18^F]Alfatide) is the first compound of this class of tracers studied in clinical cohorts. Besides the two cyclic RGD peptides c(RGDyK) bridged via a lysine, it contains a PEG moiety as pharmacokinetic modifier and a Bz-NOTA moiety for the complexation of [^18^F]AlF [304]. In nine patients with lung cancer, [^18^F]Alfatide identified all the tumors, with mean standardized uptake values of 2.90 ± 0.10. The highest activity accumulation was found in the kidneys and bladder, being a hallmark of renal clearance. The liver, spleen, and intestines also displayed moderate uptakes. Nonetheless, PET scanning with 18F-alfatide allowed for the specific imaging of α_v_β_3_ expression with superior contrast in lung cancer patients. Immunohistochemical staining confirmed α_v_β_3_ expression in tumor cells and neovasculature in SCC patients, highlighting that [^18^F]Alfatide may be a promising radiotracer for disease detection in clinical setting [315].

Several recent reports highlighted that additional integrins, such as α5β1 and αvβ6, may also be exploited for gaining mechanistic insights concerning tumorigenesis in vitro and in vivo [271]. Integrin α5β1 was documented to play critical roles in angiogenesis, while integrin αvβ6 does not seem to be involved in angiogenesis but was found highly expressed on a variety of tumors, affecting disease outcomes [271]. Although current scientific efforts seek to find the most suitable peptide structures and radiolabeling strategies, the available tracers for marking these two integrins still lack metabolic stability and show increased accumulation in normal organs such as the kidneys [304]. Therefore, further optimizations are needed to find suitable compounds for the noninvasive imaging of α5β1/αvβ6 integrins in cancer patients.

Imaging integrin α_v_β_3_ expression remains one of the most effervescent fields of molecular imaging within recent years. Currently, ^18^F-FDG PET targeting this integrin plays an increasing role in the diagnosis and management planning of head and neck SCC [316,317]. Many other RGD-derived compounds, such as ^68^Ga-NOTA-BBN-RGD (NCT02747290) [318], ^68^Ga-NOTA-3PTATE-RGD (NCT02817945) [319], and ^18^F-RGD-K5 (NCT02490891) [320], are also tested in clinical trials for their diagnostic efficiency and safety in imaging integrins in different types of cancer. Since integrin expression is not restricted to a particular cancer type, being ubiquitous in prostate, breast, brain, skin, and lung tumors, these integrin α_v_β_3_-targeted agents can broadly be applied to cancer patient management in general. Therefore, more consistent research efforts and resources should be devoted to moving the most promising α_v_β_3_ integrin-targeting agents into clinical practice in a timely manner, as they may increase the chances of an effective early diagnosis and treatment and hence reduce the burden of cancer worldwide [271].

#### 3.3.3. The Non-Invasive Visualization of Fibroblast Activation Protein (FAP)

FAP is a serine protease notorious for its heightened expression in tumor stroma [321]. As fibroblasts are highly prevalent within tumors, targeting FAP would be a valuable approach for imaging tumors and for the development of novel radiopharmaceuticals for cancer therapy [21].

The main drawback of FAP-targeting ligands is that they also portray other pathological states linked with fibroblast activation, such as lung fibrosis, liver fibrosis, atherosclerosis, or arthritis [322]. In 2017, Lindner et al. described the synthesis of at least 15 quinoline-based theranostic ligands for the targeting of FAP in cancer tissues. Of 15 synthesized FAPIs, FAPI-04 was identified as the most promising tracer in terms of binding, internalization, and preclinical pharmacokinetics [323]. The team further conjugated this molecule to ^68^Ga (^68^Ga-FAPI-04) and used the radiotracer for the first time in clinical studies for imaging two patients with metastatic breast cancer. PET/CT scans with ^68^Ga-FAPI-04 in metastasized breast cancer patients revealed high tracer uptake in metastases, highlighting that FAPI-04 represents a promising tracer for both diagnostic imaging and, possibly, targeted therapy of malignant tumors with a high content of activated fibroblasts, such as breast cancer. Finally, the same research group conjugated FAPI-04 to ^90^Y (^90^Y-FAPI-04) and administered this compound, which resulted in a reduction in pain symptoms associated with metastasis [323]. More recently, several research groups assessed ^68^Ga-FAPI PET/CT diagnostic efficiency in melanoma tumors. In a case report of liver metastasis from malignant melanoma, the lesion showed high radiotracer uptake in ^18^F-FDG PET/CT imaging; additionally, mild uptake due to osteoarthritis was observed in both knees [268]. In contrast, ^68^Ga-FAPI-04 showed lower uptake in liver lesions and, hence, showed a more prominent uptake in both knee joints compared with ^18^F-FDG, suggesting that ^18^F-FDG PET/CT may be more specific in detecting melanoma metastases than ^68^Ga-FAPI-04 when they co-exist with other non-cancerous states linked to fibroblast activation [268]. In addition, ^68^Ga-FAPI PET/CT proved to be effective in detecting a melanoma tumor in the right nasal cavity of a 56-year-old man who presented with a 1-month history of recurrent right-sided epistaxis [324]. The researchers further performed a tissue biopsy that confirmed the occurrence of the melanoma tumor, in order to eliminate the doubts of any disease associated with fibroblast activation [324]. In addition, a study conducted by Kratochwil et al. showed that 28 of the most prevalent cancers presented with remarkably high uptake and image contrast on ^68^Ga-FAPI PET/CT, suggesting that novel PET radiotracers targeting FAP may open up new applications for noninvasive tumor characterization, staging examinations, or radioligand therapy in cancer patients [325].

As a consequence of promising results showed in clinical research studies, FAP-targeted imaging agents are currently evaluated in clinical trials for their efficiency and biosafety for the diagnosis of primary and metastatic lesions in various types of cancer. One such clinical trial (NCT04441606), which was recently completed, focused on malignancies known to show variable avidity to ^18^FDG and at times, no uptake at all, such as exocrine pancreatic cancer, gastric carcinoma, mucin-producing, or Signet-ring carcinoma. The study incorporated approximately 50 participants, with the goal of comparing ^68^Ga-FAPI-04 PET efficiency to that of ^18^FDG-PET in evaluating disease burden [326]. Another ongoing clinical trial (NCT04499365) plans to enroll 500 subjects for ^68^Ga-DOTA/NOTA-FAPI-04 PET to provide robust clinical evidence regarding the use of FAP-targeted imaging for the diagnosis of primary and metastatic lesions in various types of cancer [327]. The preliminary results of this study highlighted the potential of ^68^Ga-DOTA-FAPI-04 PET/CT for the detection of the radioiodine-refractory differentiated thyroid cancer (RR-DTC) lesions and hence emphasized that FAPI tumor uptake may provide a potential therapeutic target for RR-DTC [328]. Several FAP-targeted radiotherapies are also under investigation in clinical studies (NCT05030597, NCT05506566, and NCT05543317) [329,330,331]. Nonetheless, given the ubiquitous presence of fibroblasts in cancer and the avidity of FAP-targeted imaging agents, FAP-targeted PET imaging has the potential to revolutionize the landscape of oncological PET/CT imaging and improve the clinical care of this highly problematic disease.

#### 3.3.4. Imaging the Melanocortin-1 Receptor (MC1R)

MC1R is an important melanoma-specific target that has been proposed for molecular imaging applications. At least five melanocortin receptors (MC1R-MC5R) have been described to date [332]. MC1R, which was found expressed in more than 80% of human metastatic melanomas, remains the best characterized melanocortin receptor [333]. MC1R activation is an important process that regulates melanocyte cell division. The α-melanocyte-stimulating hormone (α-MSH) binds to MC1R and activates adenylate cyclase (AC), increasing the intracellular concentration of the secondary messenger, cAMP. Subsequently, cAMP activates PKA, which phosphorylates CREB, leading to the increased expression of MITF in melanocytes. Finally, MITF induction leads to the activation of the RAS/RAF/MEK/ERK-oncogenic pathway, which promotes melanocyte cell division and survival [334]. As a consequence of a better comprehension of the α-MSH/MC1R pathway, a variety of radiolabeled α-MSH peptide analogues with MC1R binding affinities have been developed as melanoma-specific imaging probes for lesion detection. As native α-MSH (a linear tridecapeptide: Ac-Ser-Tyr-Ser-Met-Glu-His-Phe-Arg-Trp-Gly-Lys-Pro-Val-NH_2_) has a biological half-life of less than 3 min in vivo, considerable efforts have been devoted within recent decades to synthetize several modified analogs with improved biological stability and target specificity [335]. One of these synthetic peptides is ^125^I-[Nle4, D-Phe7 ]-α-MSH, often referred to as a “gold” standard in MC1R imaging due to its sub-nanomolar receptor binding affinity [336]. Moreover, in the strive to identify more potent agents for MC1R imaging, researchers radiolabeled various α-MSH peptides with an increased number of radionuclides including ^18^F, ^99^mTc, ^111^In, ^125^I, ^67^Ga, ^68^Ga, ^86^Y, and ^64^Cu, with some of them with radiotherapeutic applications [270]. Hence, the discovery that His^6^-Phe^7^-Arg^8^-Trp^9^ is the “essential core” of native α-MSH peptide facilitated the development of novel linear and cyclized α-MSH such as (DOTA-)ReCCMSH [337], NAPamide [338], DOTA-NAPamide [339,340], MTII [341], DOTA-CycMSH [342], and DOTA-Nle-CycMSH_hex_ [343,344], which showed promising results in experimental studies.

ReCCMSH, which has nanomolar affinity for MC1R, proved increased diagnostic efficacy in melanoma tumors upon labeling with PET isotopes ^64^Cu and ^86^Y. Biodistribution studies highlighted that the ^86^Y-DOTA-ReCCMSH and ^64^Cu-DOTA-ReCCMSH uptake was two times higher in tumors compared to that of the metabolic agent ^18^F-FDG. Moreover, the administration of the chelator agent CBTE2A decreased the accumulation of the 64Cu-CBTE2A-ReCCMSH peptide in normal organs such as liver, lung, heart, and spleen when compared to the administration of the parental peptide [345]. ^68^Ga-DOTA-ReCCMSH also showed promising imaging capabilities in melanoma diagnosis and its co-administration with D-lysine resulted in a significantly reduced kidney retention [346]. It is also worth mentioning that ^111^In-DOTA-GlyGlu-CycMSH, a radiotracer designed to target MC1 receptors in primary and metastatic melanoma lesions, has recently undergone a series of chemical modifications that aimed to improve its biological stability and specificity. Particularly for this compound, it was shown that the reduction in the ring size, substitution of DOTA with NOTA, L-lysine co-injection, introduction of a -GG-linker, and ^99^mTc radiolabeling may increase the melanoma tumor uptake while reducing the nonspecific kidney and liver uptake [347,348,349].

Another class of MC1-R targeting agents extensively used to detect melanomas or evaluate the cellular levels of MCR1 are the NAPamide analogs. Distinct NAPamide compounds showed different tumor penetration patterns, highlighting that certain chemical modifications may be mandatory to improve the uptake rates and imaging performances of PET tracers. For instance, ^64^Cu–DOTA–NAPamide showed a mild tumor uptake in B16/F10 xenografted melanoma, whereas ^64^Cu-NOTA-GGNle-CycMSHhex displayed an elevated uptake at 2 h after injection. These observations were of great help in fine-tuning the properties of ^64^Cu-DOTA-GGNle-CycMSHhex, as the substitution of DOTA with NOTA considerably increased the melanoma uptake and decreased the renal and liver uptake of ^64^Cu-NOTA-GGNle-CycMSHhex [350].

However, the translation of radiolabeled peptides, including α-MSH peptides in clinical practice as novel cancer imaging or therapeutic agents remains challenging because this process is laborious and relatively expensive and, most of the time, the results are modest [270]. The major hurdle in this process is the investigation and optimization of toxicological effects in a preclinical setting, which is essential for translating a compound from bench to bedside [351]. However, although many of the studies described above suggested that developed α-MSH peptides have the potential for metastatic melanoma imaging or peptide receptor-targeted radionuclide therapy, none of those agents have yet reached the clinical stage, remaining at best in preclinical assessment [270]. With so many different agents developed for the same molecular target, choosing the right agent for clinical evaluation is difficult. In addition, evaluating several agents of similar characteristics in the clinic is also impractical due to the prohibitive cost [351].

Nonetheless, the newly introduced AI approaches can address the issues associated with the synthesis, selection, and clinical validation of radiopharmaceuticals, reinforcing the advancement of precision medicine in oncology [351]. With an increasing amount of data on the structure and function of biological targets, AI can expedite radiopharmaceutical design research resulting in more rapid incorporation of these compounds in routine medical practice [352]. Computational models can offer mechanistic insights on the target-binding affinity of compounds, their absorption, metabolism, toxicity, and excretion. Therefore, in silico methods could facilitate a faster and more cost-effective design and testing of new radiotracers and radiopharmaceuticals in vitro and in vivo, reducing the need for animal models to evaluate the properties of these compounds [352]. It is also worth noting that computational modeling is not a replacement for in-lab experiments but rather a companion tool aimed to facilitate radiopharmaceutical development, as currently, no modeling can perfectly reproduce the complexity of the human body [352].

## 4. Discussion

The incidence of malignant skin tumors, especially cutaneous melanoma (CM), basal cell carcinoma (BCC), and squamous cell carcinoma (SCC), has been alarmingly increasing over the last decades [353]. Nonmelanoma skin cancers (NMSC), including BCC and SCC, are the most prevalent malignancies affecting light-skinned individuals worldwide and represent almost 95% of all cutaneous cancer diagnoses [354]. The World Health Organization reported an increase of more than 2–3 million new cases of NMSC per year, of which almost 80% are BCCs. NMSCs are generally curable but can result in considerable morbidity and mortality when treatment is delayed or inappropriate [355,356,357]. Meanwhile, the prevalence of CM is also increasing worldwide. CM accounts for less than 5% of cutaneous neoplasms and is responsible for the majority of skin cancer-associated deaths. Nonetheless, both NMSC and CM have an excellent prognosis when they are diagnosed and treated early. Accordingly, considerable research efforts should be devoted to achieving the early detection and a better understanding of the disease to reverse the progressive trend of rising incidence and mortality, especially regarding melanoma [4].

The histological examination of biopsies remains the gold standard for revealing pathological changes in tissue. However, it is a highly invasive procedure and requires sample preparation and fixation, which can alter the biological properties of the sample [358]. Emerging anatomical and molecular techniques are now available or under research to meet the need for non-invasive in vivo measurements of skin. The most popular anatomic techniques include confocal laser scanning microscopy (CLSM) [359], optical coherence tomography (OCT) [90], high-frequency ultrasound (HFUS) [117], terahertz pulsed imaging (TPI) [29], and magnetic resonance imaging (MRI) [156]. However, anatomical imaging alone may also be unsatisfactory in guiding skin cancer diagnosis and therapeutic decisions, calling for even more refined approaches in the field. Within the last decade, molecular imaging techniques such as single photon emission computed tomography (SPECT/CT) and positron emission tomography (PET) have been extensively investigated to provide additional diagnostic information and help identify specific disease mechanisms in tumors, which consolidated their position in the diagnostic strategy of human cutaneous tumors [349]. The signaling pathways and molecular targets that may be harnessed for molecular imaging in cancer are glucose metabolism [255], integrin α_v_β_3_ [271], melanocortin-1 receptor [332], PD-1/PD-L1 axis [272], and hence several other biochemical and molecular markers related to immune response and angiogenesis reside in the area of preclinical development [360]. Although both imaging modalities may be used to assess all skin cancer subtypes, anatomical imaging may be more relevant to NMSC. In contrast, molecular imaging is more appropriate for evaluating melanoma and metastasis. As almost all molecular imaging techniques rely on the use of a molecular probe, which should be approved for clinical use, it is no wonder that most of the clinical data for molecular imaging of skin cancer are on ^18F^-FDG PET imaging, as ^18^F-FDG is the only PET agent that has gained FDA approval for cancer imaging [17,361]. Finally, although the advances we witnessed in recent years in imaging techniques were groundbreaking, each method has strengths and limitations that must be fully elucidated and understood before employing them in the clinical setting.

Fluorescence microscopy is a widely used anatomical technique for the “real-time” pathological examination of freshly excised or frozen BCC specimens for diagnostic purposes and tumor margin assessment in Mohs surgery [77]. However, extensive studies are further needed to improve the FCM staining and digital staining algorithms, reduce the artifacts, and improve the turnaround time and costs associated with such a procedure [80]. RCM—another variant of CLSM—is a high-resolution imaging technique with remarkable labeling capabilities but limited penetration depth [16]. RCM enables the repetitive examination of the same skin area without causing any damage and proved useful in monitoring disease progression, tumor-associated vasculature, and inflammation in skin cancer patients [362]. Furthermore, RCM allows for the assessment of the entire lesion and may be employed to guide biopsies and define tumor margins before surgical excision [66] or other types of therapies [68]. Finally, RCM diagnostic features may help differentiate between the various histologic subtypes of skin tumors and guide the therapeutic decisions in the affected patients [71]. Although the resolution of OCT is lower than that of RCM, OCT can be used to evaluate more profound depths than RCM [87]. The most significant applications of OCT in dermatology have thus far been in diagnosing, delineating, and treatment monitoring of NMSC, especially BCCs [86]. It is worth mentioning that the hyperkeratotic epidermis of the SCC lesions prevents OCT from obtaining insights into deeper skin layers and observing the dermal–epidermal junction, which causes this technique to be inappropriate for SCC diagnosis. In addition, pigmented lesions continue to pose significant challenges in OCT imaging and, in diagnosing CM, OCT is not as accurate as dermoscopy or RCM [86]. HFUS is another method that can be used to assess the morphological changes in the skin. Ultrasound is widely used for the evaluation, staging, and follow up of patients with melanoma, as it is a non-invasive and affordable imaging method and shows increased sensitivity and specificity in detecting melanoma metastases larger than 4.5 mm in diameter that are localized deeper in soft tissues [28,130]. Concerning NMSC, HFUS may provide valuable information regarding the tumor size and depth of invasion, which is of great importance when planning the extent of surgery in the clinical management of BCC and SCC [121,123]. However, due to the inflammation and the hyperkeratotic characteristic of some SCC, HFUS may have a decreased accuracy in investigating the features of SCCs or assessing their depth of invasion. The overall low resolution, lack of functional contrast, and image quality are other limitations of HFUS [122,125].

Near-infrared (NIR) imaging using indocyanine green (ICG) is another commonly used method for skin measurement. In particular, for skin cancers, NIR fluorescence imaging proved helpful for the identification of sentinel lymph nodes (SLNs) [184] and for the visualization of fluorescence-labeled immune cells, especially to track and monitor them in living organisms during immunotherapy [185]. However, NIR imaging usually results in complex spectra that are challenging to interpret [363]. THz sensing is another anatomical technique that may help guide the non-invasive diagnosis of skin cancer and assess the therapeutic responses in the affected patients. Given the increased sensitivity of THz radiation to water and its increased penetration depth into skin and tissues, THz sensing allows for a clear demarcation of healthy and pathological tissues [134]. Although the diagnostic accuracy of THz imaging is quite satisfactory, it does not have the ability to discriminate between different skin cancer subtypes [142]. Compared to OCT, which mostly reveals the structure and morphology, THz imaging is more sensitive to the structural and chemical properties of normal and pathological tissues. Therefore, THz imaging is a valuable technique for quantitative in vivo skin analysis, which could lay the ground for a more effective diagnosis of skin lesions and pathological processes [29]. Another important optical technique is MRI, which is employed to differentiate between the different forms of skin cancers and help in the localization and delineation of that tumors that may be difficult to assess because of their topography [156,157]. MRI also allows determining the degree of invasion of malignant tumors within deeper soft tissues and hence measure their size and thickness [156,158,159]. Because of its increased sensitivity and superior soft tissue contrast, MRI is preferred over CT in patients suspected of perineural disease or deep soft tissue involvement [156]. Given the good diagnostic performances achieved in the detection of micrometastases, the Dermatologic Cooperative Oncology Group and the updated Swiss Guidelines suggested the use of the whole body-MRI as an alternative to 18-fluorodeoxyglucose (FDG) PET/CT for the staging of high-risk and metastatic (stage III or IV) melanoma and the follow-up of stage IIC or higher CM patients [162]. Although more expensive than CT, MRI provides many advantages, including a higher field of view, spatial resolution under 100 μm, and an excellent contrast, all of them without utilizing harmful ionizing radiations [162,163]. MRI scans have also been reported to aid in the early detection of brain metastases long before the first occurrence of neurological symptoms in patients with metastatic melanoma [169]. Other applications of MRI techniques in dermato-oncology may include the assessment of therapeutic responses in CM and SCC [173,174], the differentiation of pseudoprogression from progressive disease [175], and the in vivo assessment of tumor angiogenesis for tumor characterization and treatment planning [176].

Regarding molecular imaging techniques, they are mainly used for melanoma staging and follow-up. Melanoma staging relies on clinical and pathological data incorporated in the American Joint Committee on Cancer (AJCC) staging system. According to this model, routine imaging is not generally recommended in patients with lower risk (stage I and II) when specific signs or symptoms are absent. However, for clinically node-negative patients, an accurate evaluation of regional lymph nodes should be obtained by employing lymphoscintigraphy (LS) and sentinel lymph node biopsy (SLNB), which remain the gold standards of regional lymph node staging [126]. Among all the molecular imaging techniques, ^18^F-FDG PET/CT remains the procedure with the most diverse and promising applications.^18^F-FDG is the only PET radioactive tracer that received FDA approval in the 1990s and subsequent reimbursement by the Centers for Medicare and Medicaid Services (CMS); since then, its use for imaging applications in oncology has grown steadily [364]. ^18^F-FDG PET/CT plays a limited role in staging early melanoma (AJCC stages I and II); however, it is more useful in the initial staging of advanced cutaneous melanoma (AJCC stages III and IV) and the assessment of disease recurrence, demonstrating a high accuracy in the detection of lymph node, soft tissue, and visceral metastases [241]. However, increasing evidence suggests that PET/CT may not be sensitive enough in assessing melanoma brain metastases, which are one of the most life-threatening complications of the disease. PET technology allows not only for qualitative image interpretation but also for the calculation of several quantitative parameters related to metabolic activity, such as maximum and mean standardized uptake value (SUVmax and SUVmean) and metabolic volume, such as metabolic tumor volume (MTV) and total lesion glycolysis (TLG), which may be indicative of disease progression and clinical outcome in CM patients [252,254]. More recently, ^18^F-FDG PET/CT has become part of the latest immunotherapy response criteria, such as the PET Response Evaluation Criteria for Immunotherapy (PERCIMT) and Immunotherapy-modified PET Response Criteria in Solid Tumors (imPERCIST) criteria, showing promising results in predicting responses to treatment and patient survival in CM patients (Table 2) [260]. Yet, PET-CT has the potential to differentiate between pseudoprogression and real progression, which is essential for proper clinical management in the affected patients (Table 2) [263]. Although FDG-PET may offer valuable information on primary staging, response assessment post-therapy, staging of suspected recurrence, and prognosis, the results of PET examinations should be interpreted with care, as there are a lot of aspects that may impact their accuracy. Imaging pitfalls are considerably affected by several factors, including patient preparation and lesion characteristics as well as the preferential bio-distribution of ^18^F-FDG in certain tissues or organs. Pitfalls dramatically impact scan sensitivity and specificity, as they may result in false positives or negatives [241]. False-positive findings may result in unnecessary and invasive procedures; yet, false-negative scans may be associated with delayed diagnosis and treatment, thereby dramatically affecting the patient and clinician [242].

Although most imaging techniques show promising results in clinical cohorts, numerous aspects must be investigated before validating them as robust diagnostic instruments. Almost all the described optical methods have limitations, such as high cost, the need for specialized medical personnel, low specificity, and reduced efficiency when employed singularly in the clinical setting [365,366]. Another issue to consider is that most are not readily applicable to lesions with hyperkeratosis and inflammation or lesions located in unconventional anatomic sites [241,242]. One possibility that has been suggested to overcome the limited sensitivity and specificity values is the combined use of optical modalities, which may result in performances larger than those achieved by the sole use of one of the technologies. For instance, the combined use of RCM and OCT within the same device delivers enhanced capabilities for skin cancer diagnosis and especially for therapy guidance [367]. In addition, the information obtained with combined PET/CT delivers both the metabolic data from FDG-PET and the anatomic information from CT in a single examination. The information obtained by PET/CT is more accurate in evaluating patients with known or suspected malignancy than either PET or CT alone or PET and CT received separately but interpreted together [242]. Furthermore, using these techniques combined with tools such as learning algorithms, namely machine learning and deep learning, can improve disease management in the clinical setting. However, despite all the enthusiasm generated by the application of AI in imaging interpretation, the clinician remains the person who must decide which lesion among thousands and which patient would benefit most from further testing [368]. Therefore, although AI tools can facilitate and enhance human work, they are not yet able to replace the work of physicians and other healthcare staff as such. Using such devices on non-preselected lesions can lead to erroneous diagnoses and compromise the treatment for various pathologies [369].

In the last two years, in which the COVID-19 pandemic restricted cancer patients’ access to healthcare facilities, teledermatology has gained increased popularity among patients and clinicians [370]. Indeed, optical techniques for skin cancer diagnosis have the potential to be part of teledermatology for primary care in remote areas. Moreover, optical devices may be smoothly coupled with AI, which ensures they gain improved performances and rapid dissemination in social environments [16]. Nonetheless, to prevent those emerging techniques from remaining only at the stage of an exploratory project, more future large, multicenter, and prospective clinical trials regarding the utility of these techniques should be conducted. The healthcare providers, industry experts, patients, and communities of each country should join hands to support such initiatives, which have the potential to catalyze the adoption of optical technologies into clinical practice and reduce the socio-economic burden of cancer worldwide.

## Figures and Tables

**Figure 1 ijms-24-01079-f001:**
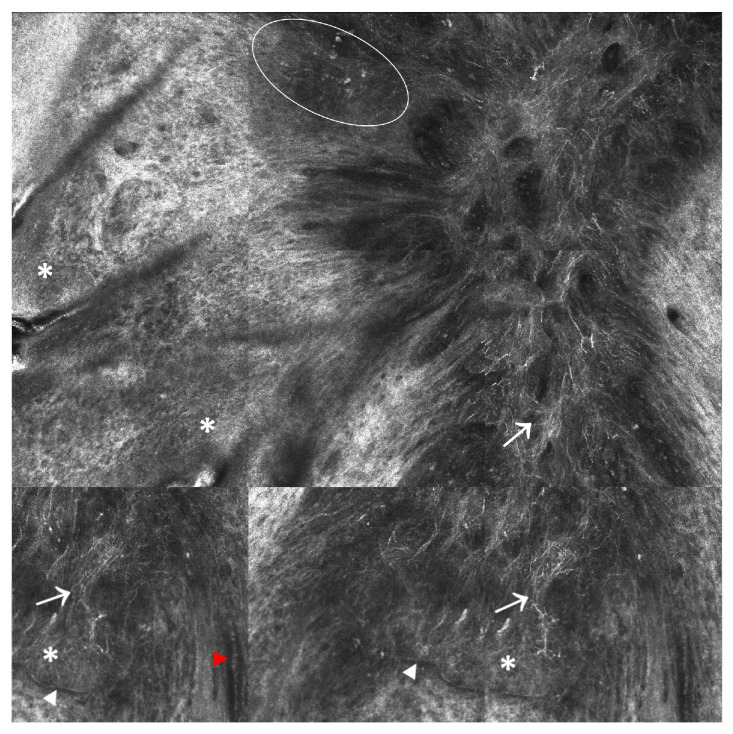
Basal cell carcinoma. RCM mosaic (1500 × 1500 µm^2^) showing elongated cord-like structures and tumor islands (*) of different sizes delineated by dark clefts (white arrowhead). Large canalicular blood vessels (red arrowhead). Sparse plump-bright cells representing melanophages (white circle) and numerous bright cells with thin dendritic structures corresponding to melanocytes (white arrows) within the tumor islands.

**Figure 2 ijms-24-01079-f002:**
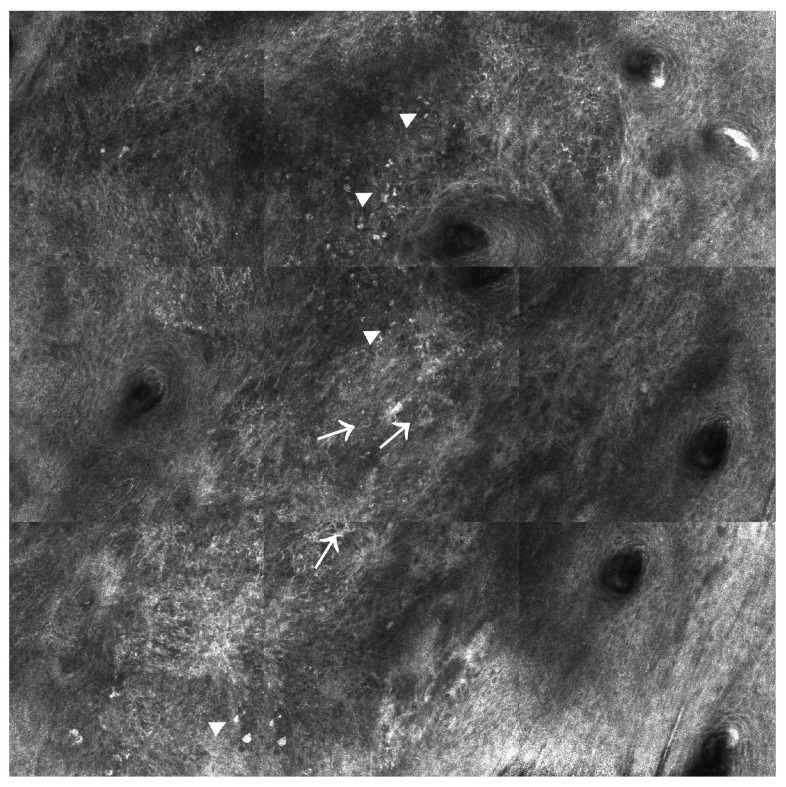
Squamous cell carcinoma. RCM mosaic (1500 × 1500 µm^2^) showing atypical honeycomb pattern with broadened and irregular intercellular connections, pleomorphic keratinocytes, varying in size and shape and areas of total architectural disarray; scarce large, round, nucleated cells representing dyskeratotic cells (white arrows) and scattered small, bright inflammatory cells (white arrowheads).

**Figure 3 ijms-24-01079-f003:**
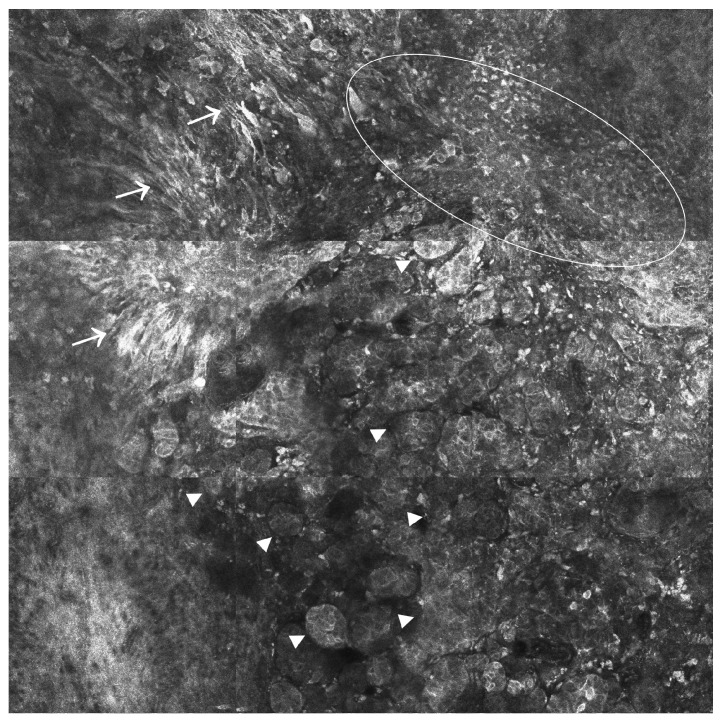
Cutaneous melanoma. RCM mosaic (1500 × 1500 µm^2^) showing severe dermo-epidermal junction disarray and pleomorphic bright cells: large dendritic atypical cells (white arrows) with the tendency to form aggregates and roundish cells of different sizes in some areas forming scattered irregular clusters (white arrowheads), in others distributed in a diffuse pattern (white circle).

**Table 1 ijms-24-01079-t001:** Comparison of different imaging techniques available for skin cancer imaging ^1^.

Technique	Spectra	Contrast Agent Needed	Resolution	Penetration Depth	Advantages	Limitations
RCM	Near-infrared	No	0.5–1 µm [34]	200 µm [16]	High resolution; fast in real-time imaging of the tumor-associated vasculature and inflammation; tumor margins assessment before surgical excision [33]	Limited penetration, which may result in false-negative results for tumors below the papillary dermis; extensive training [16]; backscattered nuclear contrast [78]
FCM	Near-infrared	Yes	0.4–1 µm [77]	200 µm [16]	Examination of freshly excised or frozen specimens for diagnostic purposes; tumor margin assessment in Mohs surgery; increased resolution [77]	Difficulties in preserving tissue properties when preparing it for FCM examination [78]
MPM (TPEF)	Near-infrared	Sometimes	>300 μm [233]	150–200 μm in the skin; 2 mm in the brain [234]	High resolution; suitable for ex vivo clinical investigations; enable the assessment of size variation of cell nuclei, blood vessel architecture, and inflammatory states [235]	Imaging resolution and depth dependable on the composition of the tissues of interest; varying indices of refraction distort the excitation focus, resulting in signal loss and a reduction in image resolution; local photodamage [234]
OCT	Infrared	No	2–20 µm [86]	2–3 mm [16]	High resolution; ideal for cross-sectional imaging [16]	Limited penetrance depth; hard to distinguish SCC and AK from scar and inflammation [86]; false negative rates in thin melanomas and false positive rates in dysplastic naevi [108]
RCM and OCT	Infrared	No	3–20 µm [236]	200–1000 µm [236]	Comprehensive 3D sampling in vivo; increased diagnostic accuracy [78]; tumor margin assessment in BCCs [236]	Complexity of the data processing software [236]
HFUS	20–30 MHz	Sometimes	50–500 µm [16]	6–7 mm [16]	Wide availability; enable the assessment of vascularization and therapeutic responses in BCCs; suitable for SLN visualization in tumors > 4.5 mm [130]	Low resolution; lack of functional contrast; inability to detect melanoma micrometastases [131]
TPI	Terahertz wave	No	20–200 µm [17]	Frequency dependent, but less than 1 mm [17]	Allows a clear demarcation of healthy and pathological tissues based on their water content [134]; potential therapeutic applications in melanoma [30]	Limited availability; strong water absorption, which limits the penetration depth in fresh tissues; not fully validated for use in clinical settings [136]
MRI	Radio waves	Sometimes	25–100 µm [17]	No limit [17]	High accuracy; good tissue contrast; asymptomatic metastases detection before the occurrence of neurological symptoms; assessment of intratumoral characteristics; enable adaptative treatment strategy changes when needed [153,169]	Increased acquisition time; variability in signal intensity amongst inter-patient and intra-patient acquisitions [237]; difficulty in interpretation; expensive [17]
NIR	Near-infrared	Yes	1.2–20 µm [238]	~760 µm [239]	SLNB identification [187]; visualization of fluorescence-labeled immune cells in vivo during immunotherapy; less hazardous to biological samples than the short-wavelength excitation light in visible fluorescence bioimaging [193]	The need for assay standardization; assessment of assay reproducibility in larger clinical cohorts [186]
SPECT/CT	-	Yes, radiotracers	8–12 mm [240]	Limitless [240]	SN identification when the lymphatic drainage is unclear [221]; detection of the second lymph node level [219]; increased availability and half-life of SPECT radiopharmaceuticals; remarkable imaging abilities of the biological processes in vivo [226]	Limited resolution; increased workload for pathology departments and increased risk of cancellation of the SNB procedure on the day of surgery, which may negatively impact nodal relapse-free survival [224]
PET/CT	-	Yes, radiotracers	4–6 mm [240]	Limitless [240]	Useful in the initial staging of advanced cutaneous melanomas (AJCC stages III and IV) and assessment of disease recurrence, highly accurate in detecting the lymph node, soft tissue, and visceral metastases [241]	Limited role in staging early melanoma (AJCC stages I and II) [241]; expensive technique; prone to imaging pitfalls related to patient preparation, inflammatory conditions, and lesion characteristics [242]; cumulative exposure to ionizing radiation [21]

^1^ BCC—basal cell carcinoma; NMSC—non-melanoma skin cancer; AK—actinic keratoses; SCC—squamous cell carcinoma; HD-OCT—high-definition OCT; SNs—sentinel nodes; SNLB—sentinel lymph node biopsy; HHi—Hedgehog inhibitors; AJCC—American Joint Committee on Cancer; 3D—tridimensional.

**Table 2 ijms-24-01079-t002:** Prospective applications of anatomical and molecular imaging techniques in dermato-oncology.

Technique(s)	Suitable for the Identification of SN	Suitable for Skin Cancer Subtyping	Suitable for Patient Monitoring Following Therapy	Suitable for the Evaluation of CM Pseudo Progression
RCM	No	Yes	Yes	Not available
FCM	Yes	No	Not available	Yes
MPM	No	Yes	Not available	Not available
OCT	No	Yes	Yes	Not available
RCM/OCT	No	Not available	Yes	Not available
HFUS	Yes	Yes	Yes	Not available
TPI	No	No	Yes	Not available
MRI	Not available	Yes	Yes	Yes
NIR	Yes	No	Yes	Yes
SPECT/CT	Yes	Not available	Not available	Not available
PET/CT	Yes	No	Yes	Yes

## Data Availability

Not applicable.

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
