# Peer review of "Skin Cancer Pathobiology at a Glance: A Focus on Imaging Techniques and Their Potential for Improved Diagnosis and Surveillance in Clinical Cohorts"

_ijms, 2023, doi:10.3390/ijms24021079_

Round 1

Reviewer 1 Report

1. would recommend adding a section on duel photo (multiphoton) microscopy

2. In the abstract the authors write "Optical techniques proved to be promising companion tools to invasive histopathological approaches in evaluating tissue morphology." How so? They are definitely companion tools for clinicians but I am not sure they are companion tools for histopathology.  

3.  In the introduction the authors mention the potential benefits of imaging technologies but fail to mention the potential harms such as overdiagnosis and false diagnosis (false positives and false negatives). 

4. In the introduction the authors state that visual inspection is inaccurate and depends on dermatologist training. I agree with this but the same is true for the imaging instruments they discuss. For example, the accuracy of RCM, OCT, US are not only reader-dependent but also operator-dependent! Thus, image interpretation also has large inter-observer and intra-observer variability.  

5. Despite all imaging instruments and the application of AI in their interpretation, the clinician remains the person who must decide which lesion among thousands and which patient would benefit most from further testing. I highlight this because the authors paint a poor picture of the clinician and make the point that imaging devices can improve the situation. I strongly disagree with this view. These imaging devices will prove beneficial only in the hands of good clinicians! Using these devices on non-pre-selected lesions will be a disaster for the care of patients.

6.  In the RCM section the authors again mention that RCM is an important adjunct to histopath. How so? The way I see it is that RCM is another level of in-vivo morphology (clinical, dermoscopy, RCM, etc) that provides a bridge between the clinical and histopathology but it does not replace it nor is it an adjunct. RCM can help in margin mapping and providing insights into directing the clinician on where to biopsy to make sure the most representative area is sampled. But in all scenarios, the hisopath remains the standard.  

7. For completeness, the authors should be aware of work to apply AI to RCM image interpretation. Also, there are papers showing that gray-scale RCM images can be converted into H&E like color images.

8. line 531: awkward sentence structure. 

9. line 1304. the word "planar" in this sentence is incorrect.

10. In table 1 the authors write that HFUS is operator-dependant. Not sure why this is specifically selected out from the others (RCM, OCT, etc) since they are all operator and reader dependant!

Author Response

Dear Reviewer,

We want to thank you for the thorough reading of this manuscript and your thoughtful recommendations, which greatly help revise the manuscript.

We agree with your comments and have revised our paper accordingly. 

Q1: would recommend adding a section on duel photo (multiphoton) microscopy

A1: We appreciate the reviewer’s comment. As suggested, we have included a section on multiphoton microscopy in the first part of our manuscript, lines 555-669.

 Q2: In the abstract the authors write "Optical techniques proved to be promising companion tools to invasive histopathological approaches in evaluating tissue morphology." How so? They are definitely companion tools for clinicians but I am not sure they are companion tools for histopathology. 

 A2: We are thankful for this suggestion. We decided to delete this phrase because it can create confusion from the scientific point of view. Also, we found that some of its content was redundant with the previous sentence (lines 31-32).

Q3: In the introduction the authors mention the potential benefits of imaging technologies but fail to mention the potential harms such as overdiagnosis and false diagnosis (false positives and false negatives).

A3: We agree with the reviewer’s comment. We have included the potential harms of imaging techniques in the introduction of our manuscript, lines 113-133.

Q4: In the introduction the authors state that visual inspection is inaccurate and depends on dermatologist training. I agree with this but the same is true for the imaging instruments they discuss. For example, the accuracy of RCM, OCT, US are not only reader-dependent but also operator-dependent! Thus, image interpretation also has large inter-observer and intra-observer variability. 

A4: We included this idea in the introduction of our manuscript, lines 129-133.

Q5: Despite all imaging instruments and the application of AI in their interpretation, the clinician remains the person who must decide which lesion among thousands and which patient would benefit most from further testing. I highlight this because the authors paint a poor picture of the clinician and make the point that imaging devices can improve the situation. I strongly disagree with this view. These imaging devices will prove beneficial only in the hands of good clinicians! Using these devices on non-pre-selected lesions will be a disaster for the care of patients.

A5: We are thankful to the reviewer for pointing out this misunderstanding. We have added some important considerations regarding the critical role of the clinician in screening malignant skin neoplasms, even in the context of highly performant AI devices (lines 2375- 2383).

Q6: In the RCM section the authors again mention that RCM is an important adjunct to histopath. How so? The way I see it is that RCM is another level of in-vivo morphology (clinical, dermoscopy, RCM, etc) that provides a bridge between the clinical and histopathology but it does not replace it nor is it an adjunct. RCM can help in margin mapping and providing insights into directing the clinician on where to biopsy to make sure the most representative area is sampled. But in all scenarios, the hisopath remains the standard. 

A6: The authors thank you for the suggestion. We have corrected it in the text: lines 190-193 and lines 359-374.

Q7: For completeness, the authors should be aware of work to apply AI to RCM image interpretation. Also, there are papers showing that gray-scale RCM images can be converted into H&E like color images.

A7: We have added it into the manuscript; you can find it at lines 378-391.

Q8: line 531: awkward sentence structure.

A8: Line 531 became line 729; we have revised the sentence and improved its readability. 

Q9:  line 1304. the word "planar" in this sentence is incorrect.

A9: Line 1304 became line 1546;

We are thankful to the reviewer for the observation; we checked, however, and the term used in the cited publication is "planar" (Perissinotti et al., 2018).

Q10: In table 1 the authors write that HFUS is operator-dependant. Not sure why this is specifically selected out from the others (RCM, OCT, etc) since they are all operator and reader dependant!

A10: We agree with the reviewer; we removed this erroneous observation from the table.

We hope that you find our responses satisfactory and that the manuscript is now acceptable for publication.       

Kind regards,

Authors of ijms-2061208

Reviewer 2 Report

This is a comprehensive, very well-organized, and detailed review of currently available imaging techniques/methods for the diagnosis of skin cancer. There are about half a dozen grammatical mistakes, which are minor and can be easily corrected. They are on lines: 582 (harness); 596 (shown); 713 (shown); 734 (hard); 745 (strong); 759 (recommend); 782 (when they had been proposed for the first time); 787 (its); 807, 812 and 813 (micron .... micron); 813 (similar to other); 849 (it does not have); 850 (TPI may be able to assist); 879 (evaluation of tumor extension); 898 (those tumors); 900 ( and hence can measure); 902 (of any size on the face); 903 (2 cm on the trunk); 916 (on the head and neck); 926 (distinction of BCCS from SCCs); 984 (the most common ones consisted of switching ...); 987 (probably due to the fact); 1050 (cyanines seem to have); 1055 (including those in dermato-oncology); 1077 (indocyanine green); 1153 (nanoparticles (....), targeted ...); 1342 (significantly higher); 1343 (recent advancies); 1359 (tissular???  - could this be tissue?); 1427 (substantially align); 1428 (who conducted); 1564 (which acts); 1785 and 1786 (lung fibrosis is mentioned twice); 1805 (cancerous states linked); and 2006 and 2007 (it does not have the ability to discriminate).

In line 1934, in my opinion, the sentence should state: "Histological examination of biopsies remains the gold standard for revealing pathological changes in tissue". The fragment "usually accompanied by optical microscopy" should be deleted. 

Thank you.

Author Response

Dear Reviewer,

Thank you for thoroughly reading this manuscript and for your thoughtful recommendations, which helped revise the manuscript.

We agree with your comments and have revised our paper accordingly.

Q1: There are about half a dozen grammatical mistakes, which are minor and can be easily corrected. They are on lines: 582 (harness); 596 (shown); 713 (shown); 734 (hard); 745 (strong); 759 (recommend); 782 (when they had been proposed for the first time); 787 (its); 807, 812 and 813 (micron .... micron); 813 (similar to other); 849 (it does not have); 850 (TPI may be able to assist); 879 (evaluation of tumor extension); 898 (those tumors); 900 ( and hence can measure); 902 (of any size on the face); 903 (2 cm on the trunk); 916 (on the head and neck); 926 (distinction of BCCS from SCCs); 984 (the most common ones consisted of switching ...); 987 (probably due to the fact); 1050 (cyanines seem to have); 1055 (including those in dermato-oncology); 1077 (indocyanine green); 1153 (nanoparticles (....)targeted ...); 1342 (significantly higher); 1343 (recent advancies); 1359 (tissular???  - could this be tissue?); 1427 (substantially align); 1428 (who conducted); 1564 (which acts); 1785 and 1786 (lung fibrosis is mentioned twice); 1805 (cancerous states linked); and 2006 and 2007 (it does not have the ability to discriminate).

A1: We are grateful to the Reviewer for these observations. We have corrected the indicated grammatical errors. Please find below the lines where these changes have been included in the revised version of the manuscript.

Line 582 became line 794 (harness);

Line 596 was deleted (shown)- The paragraph was reformulated at the suggestion of Reviewer 3;

Line 713 became line 936 (shown);

Line 734 became line 957 (hard);

Line 745 became line 968 (strong);

Line 759 became line 982 (recommend);

Line 782 became line 1005 (when they had been proposed for the first time);

Line 787 became line 1011 (its);

Lines 807, 812 and 813 became lines 1031 and 1037 (micron .... micron)- the word "micron" has been replaced with the corresponding abbreviation;

Line 813 became line 1037 (similar to other);

Line 849 became line 1074 (it does not have);

Line 850 became line 1076 (TPI may be able to assist);

Line 879 became line 1104 (evaluation of tumor extension);

Line 898 became line 1123 (those tumors);

Line 900 became line 1125 (and hence can measure);

Line 902 became line 1128 (of any size on the face);

Line 903 became line 1128 (2 cm on the trunk);

Line 916 became line 1141 (on the head and neck);

Line 926 became line 1151 (distinction of BCCS from SCCs);

Line 984 became line 1209 (the most common ones consisted of switching ...);

Line 987 became line 1212 (probably due to the fact);

Line 1050 became line 1290 (cyanines seem to have);

Line 1055 became line 1295 (including those in dermato-oncology);

Line 1077 became line 1319 (indocyanine green)- the syntagm" indocyanine green " has been replaced with the corresponding abbreviation;

Line 1153 became line 1399 (nanoparticles (....), targeted ...)-
the second appearance of the word "nanoparticles" was removed;

Line 1342 became line 1588 (significantly higher);

Line 1343 became line 1589(recent advancies);

Line 1359 became line 1605 (tissular???  - could this be tissue?)- the phrase was considered redundant and upon the observations of the third reviewer, it was eliminated;

Line 1427 became line 1677 (substantially align);

Line 1428 became line 1678 (who conducted);

Line 1564 became line 1814 (which acts);

Line 1785 and 1786 became lines 2035 and 2036 (lung fibrosis is mentioned twice);

Line 1805 became line 2055 (cancerous states linked);

Lines 2006 and 2007 became lines 2257 and 2258 (it does not have the ability to discriminate).

Q2: In line 1934, in my opinion, the sentence should state: "Histological examination of biopsies remains the gold standard for revealing pathological changes in tissue". The fragment "usually accompanied by optical microscopy" should be deleted. 

A2: Line 1934 became line 2184. We are thankful for this suggestion. We decided to delete the fragment "usually accompanied by optical microscopy" because it can create confusion from the scientific point of view.

We hope that you find our responses satisfactory and that the manuscript is now acceptable for publication.

Kind regards,

Authors of ijms-2061208

Reviewer 3 Report

The followings are some concerns and comments have been pointed out that the authors may want to consider. Thank you.

1) Line 85: Please cite the reference within the sentence. Check throughout the manuscript.

2) Line 217 Figure 1: Where are the figures come from? Literature? Please cite them.

3) Line 239: Please use italic p throughout the manuscript as it refers to a p-value.

4) Line 240: Please be consistent with or without a space before or after the signs throughout the manuscript.

5) Line 275 Figure 2: see comment on Figure 1.

6) Line 294 Figure 3: see comment on Figure 1.

7) Line 1389 Table 1: Please include references.

8) Line 2078 Table 2: I’d suggest the authors just use “Yes, No…” to make your table looks clean and cleaner.

9) Please use “µ” instead of “u”.

10) Please seriously check the formation throughout the manuscript.

11) The authors should summarize the references properly instead of the whole paragraph description. For example, lines 426-471, lines 593-618, and so on. Please seriously check throughout the manuscript and cut unnecessary words even references.

Author Response

Dear Reviewer,

We want to thank you for the thorough reading of this manuscript and your thoughtful recommendations, which greatly help revise the manuscript.

We agree with your comments and have revised our paper accordingly. 

Q1: Line 85: Please cite the reference within the sentence. Check throughout the manuscript.

A1: We are grateful to the reviewer for the observation; we have corrected and added the correct reference (line 85).

Q2:  Line 217 Figure 1: Where are the figures come from? Literature? Please cite them.

A2: Thanks to the reviewer for the observation; all three images are original and belong to our research group.

Q3: Line 239: Please use italic p throughout the manuscript as it refers to a p-value.

A3: We agree with the reviewer; we made these changes in the manuscript (lines 265, 266, 913, 916, 1177)

Q4: Line 240: Please be consistent with or without a space before or after the signs throughout the manuscript.

A4:  We are grateful to the reviewer for this information; we have made these changes in the manuscript.

Q5: Line 275 Figure 2: see comment on Figure 1.

Q6: Line 294 Figure 3: see comment on Figure 1.

A5-6: All three images are original and belong to our research group.

Q7:  Line 1389 Table 1: Please include references.

A7: Thank you so much for your comment; we have updated the table according to your recommendations.

Q8:  Line 2078 Table 2: I’d suggest the authors just use “Yes, No…” to make your table looks clean and cleaner.

A8: We agree with the reviewer and modified the table accordingly.

Q9:  Please use “µ” instead of “u”.

A9: We made the necessary changes in the description of the three images where the wrong symbol was insered.

Q10:  Please seriously check the formation throughout the manuscript.

A10: Thanks for the suggestion; we have changed the format where needed within the manuscript (e.g., lines 1183, 1184, 1281, 1582).

Q11:  The authors should summarize the references properly instead of the whole paragraph description. For example, lines 426-471, lines 593-618, and so on. Please seriously check throughout the manuscript and cut unnecessary words even references.

A11: Thank you for this suggestion that helped us refine the content of our manuscript and improve its readability. The paragraphs that underwent a significant revision are the following: lines 198-210; lines 347-352; lines 447-451; lines 480-508; lines 729-740; lines 805-840; lines 1325-1331; lines 1604-1635 and so on.

We hope that you find our responses satisfactory and that the manuscript is now acceptable for publication.       

Kind regards,

Authors of ijms-2061208

Round 2

Reviewer 3 Report

Dear Authors, thank you for the update. I don't have any further concerns now. Just please double-check and generate a clean version for publication. Good luck.

Author Response

Dear Reviewer,

Thank you for your valuable comments and feedback regarding our manuscript. Unfortunately, we couldn't generate a clean version of the manuscript as it must also be revised by a third reviewer, which should identify all the changes performed during the peer-review process herein. 

Once again, many thanks for your corrections and suggestions.

Kind regards,

Authors of ijms-2061208